# A Generalized Algorithm for Multi-Objective Reinforcement Learning and Policy Adaptation

**Runzhe Yang**
Department of Computer Science
Princeton University
runzhey@cs.princeton.edu

**Xingyuan Sun**
Department of Computer Science
Princeton University
xs5@cs.princeton.edu

**Karthik Narasimhan**
Department of Computer Science
Princeton University
karthikn@cs.princeton.edu

## Abstract

We introduce a new algorithm for multi-objective reinforcement learning (MORL) with linear preferences, with the goal of enabling few-shot adaptation to new tasks. In MORL, the aim is to learn policies over multiple competing objectives whose relative importance (*preferences*) is unknown to the agent. While this alleviates dependence on scalar reward design, the expected return of a policy can change significantly with varying preferences, making it challenging to learn a single model to produce optimal policies under different preference conditions. We propose a generalized version of the Bellman equation to learn a single parametric representation for optimal policies over the space of all possible preferences. After an initial learning phase, our agent can execute the optimal policy under any given preference, or automatically infer an underlying preference with very few samples. Experiments across four different domains demonstrate the effectiveness of our approach.[1]

## 1 Introduction

In recent years, there has been increased interest in the paradigm of multi-objective reinforcement learning (MORL), which deals with learning control policies to simultaneously optimize over several criteria. Compared to traditional RL, where the aim is to optimize for a scalar reward, the optimal policy in a multi-objective setting depends on the *relative preferences* among competing criteria. For example, consider a virtual assistant (Figure 1) that can communicate with a human to perform a specific task (e.g., provide

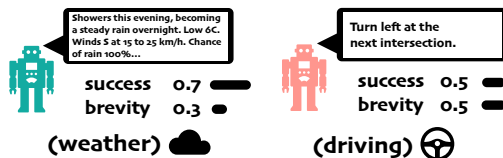

**Figure 1:** Task-oriented dialogue policy learning is a real-life example of unknown linear preference scenario. Users may expect either briefer dialogue or more informative dialogue depending on the task.

weather or navigation information). Depending on the user's relative preferences between aspects like success rate or brevity, the agent might need to follow completely different strategies. If success is all that matters (e.g., providing an accurate weather report), the agent might provide detailed responses or ask several follow-up questions. On the other hand, if brevity is crucial (e.g., while providing

turn-by-turn guidance), the agent needs to find the shortest way to complete the task. In traditional RL, this is often a fixed choice made by the designer and incorporated into the scalar reward. While this suffices in cases where we know the preferences of a task beforehand, the learned policy is limited in its applicability to scenarios with different preferences. The MORL framework provides two distinct advantages – (1) reduced dependence on scalar reward design to combine different objectives, which is both a tedious manual task and can lead to unintended consequences [1], and (2) dynamic adaptation or transfer to related tasks with different preferences.

However, learning policies over multiple preferences under the MORL setting has proven to be quite challenging, with most prior work using one of two strategies [2]. The first is to convert the multi-objective problem into a single-objective one through various techniques [3, 4, 5, 6] and use traditional RL algorithms. These methods only learn an 'average' policy over the space of preferences and cannot be tailored to be optimal for specific preferences. The second strategy is to compute a set of optimal policies that encompass the entire space of possible preferences in the domain [7, 8, 9]. The main drawback of these approaches is their lack of scalability – the challenge of representing a Pareto front (or its convex approximation) of optimal policies is handled by learning several individual policies, which can grow significantly with the size of the domain.

In this paper, we propose a novel algorithm for learning a *single policy network* that is optimized over the entire space of preferences in a domain. This allows our trained model to produce the optimal policy for any user-specified preference. We tackle two concrete challenges in MORL: (1) provide theoretical convergence results of a multi-objective version of Q-Learning for MORL with linear preferences, and (2) demonstrate effective use of deep neural networks to scale MORL to larger domains. Our algorithm is based on two key insights – (1) the optimality operator for a generalized version of Bellman equation [10] with preferences is a valid contraction, and (2) optimizing for the convex envelope of multi-objective Q-values ensures an efficient alignment between preferences and corresponding optimal policies. We use hindsight experience replay [11] to re-use transitions for learning with different sampled preferences and homotopy optimization [12] to ensure tractable learning. In addition, we also demonstrate how to use our trained model to automatically infer hidden preferences on a new task, when provided with just scalar rewards, through a combination of policy gradient and stochastic search over the preference parameters.

We perform empirical evaluation on four different domains – deep sea treasure (a popular MORL benchmark), a fruit tree navigation task, task-oriented dialog, and the video game Super Mario Bros. Our experiments demonstrate that our methods significantly outperform competitive baselines on all domains. For instance, our envelope MORL algorithm achieves an **%** improvement on average user utility compared to the scalarized MORL in the dialog task and a factor **2x** average improvement on SuperMario game with random preferences. We also demonstrate that our agent can reasonably infer hidden preferences at test time using very few sampled trajectories.

## 2   Background

A multi-objective Markov decision process (MOMDP) can be represented by the tuple $\langle \mathcal{S}, \mathcal{A}, \mathcal{P}, \boldsymbol{r}, \Omega, f_{\boldsymbol{\Omega}} \rangle$ with state space $\mathcal{S}$, action space $\mathcal{A}$, transition distribution $\mathcal{P}(s'|s, a)$, vector reward function $\boldsymbol{r}(s, a)$, the space of preferences $\Omega$, and preference functions, e.g., $f_{\boldsymbol{\omega}}(\boldsymbol{r})$ which produces a scalar *utility* using preference $\boldsymbol{\omega} \in \Omega$. In this work, we consider the class of MOMDPs with linear preference functions, i.e., $f_{\boldsymbol{\omega}}(\boldsymbol{r}(s, a)) = \boldsymbol{\omega}^{\mathsf{T}} \boldsymbol{r}(s, a)$. We observe that if $\boldsymbol{\omega}$ is fixed to a single value, this MOMDP collapses into a standard MDP. On the other hand, if we consider all possible returns from an MOMDP, we have a Pareto frontier $\mathcal{F}^* := \{ \hat{\boldsymbol{r}} \mid \nexists \hat{\boldsymbol{r}}' \geq \hat{\boldsymbol{r}} \}$, where the return $\hat{\boldsymbol{r}} := \sum_t \gamma^t \boldsymbol{r}(s_t, a_t)$. And for all possible preference in $\Omega$, we define a convex coverage set (CCS) of the Pareto frontier as:

$$ \mathrm{CCS} := \{ \hat{\boldsymbol{r}} \in \mathcal{F}^* \mid \exists \boldsymbol{\omega} \in \Omega \text{ s.t. } \boldsymbol{\omega}^{\mathsf{T}} \hat{\boldsymbol{r}} \geq \boldsymbol{\omega}^{\mathsf{T}} \hat{\boldsymbol{r}}', \forall \hat{\boldsymbol{r}}' \in \mathcal{F}^* \}, $$

which contains all returns that provide the maximum cumulative utility. Figure 2 (a) shows an example of CCS and the Pareto frontier. The CCS is a subset of the Pareto frontier (points A to H, and K), containing all the solutions on its outer convex boundary (excluding point K). When a specific linear preference $\boldsymbol{\omega}$ is given, the point within the CCS with the largest projection along the direction of the relative importance weights will be the optimal solution (Figure 2(b)).

Our goal is to train an agent to recover policies for the *entire CCS* of MOMDP and then adapt to the optimal policy for any given $\boldsymbol{\omega} \in \Omega$ at test time. We emphasize that we are not solving for a single,

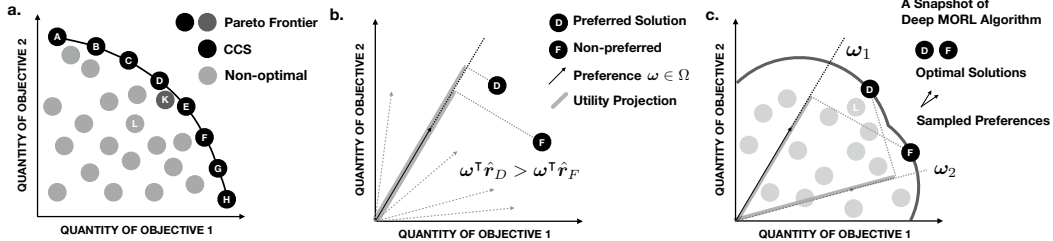

**Figure 2:** (a) The Pareto frontier may encapsulate local concave parts (points A-H, plus point K), whereas CCS is a convex subset of Pareto frontier (points A-H). Point L indicates a non-optimal solution. (b) Linear preferences select the optimal solution from CCS with the highest utility, represented by the projection length along preference vector. Arrows are different linear preferences, and points indicate possible returns. Return D has better cumulative utility than return F under the preference in solid line. (c) The scalarized MORL algorithms (e.g., [13]) find the optimal solutions at a stage while they are not aligned with preference, e.g., two optimal solutions D and F in the CCS, misaligned with preferences $\boldsymbol{\omega}_2$ and $\boldsymbol{\omega}_1$. The scalarized update cannot use the information of $\max_a Q(s, a, \boldsymbol{\omega}_1)$ (corresponding to F) to update the optimal solution aligned with $\boldsymbol{\omega}_2$ or vice versa. It only searches along $\boldsymbol{\omega}_1$ direction leading to non-optimal L, even if solution D has been seen under $\boldsymbol{\omega}_2$. It still requires many iterations for the value-preference alignment.

unknown $\boldsymbol{\omega}$, but instead aim for generalization across the entire space of preferences. Accordingly, our MORL setup has two phases:

**Learning phase.** In this phase, the agent learns a set of *optimal policies* $\Pi_{\mathcal{L}}$ corresponding to the entire CCS of the MOMDP, using interactions with the environment and historical trajectories. For each $\pi \in \Pi_{\mathcal{L}}$, there exists at least one linear preference $\boldsymbol{\omega}$ such that no other policy $\pi'$ generates higher utility under that $\boldsymbol{\omega}$:

$$\pi \in \Pi_{\mathcal{L}} \Rightarrow \exists \, \boldsymbol{\omega} \in \Omega, \text{s.t. } \forall \pi' \in \Pi, \boldsymbol{\omega}^{\mathsf{T}} \boldsymbol{v}^{\pi}(s_0) \geq \boldsymbol{\omega}^{\mathsf{T}} \boldsymbol{v}^{\pi'}(s_0),$$

where $s_0$ is a fixed initial state, and $\boldsymbol{v}^{\pi}$ is the value function, i.e., $\boldsymbol{v}^{\pi}(s) = \mathbb{E}_{\pi}[\hat{\boldsymbol{r}}|s_0 = s]$. Given any preference $\boldsymbol{\omega}$, $\Pi_{\mathcal{L}}(\boldsymbol{\omega})$ determines the optimal policy.

**Adaptation phase.** After learning, the agent is provided a new task, with either a) a preference $\boldsymbol{\omega}$ specified by a human, or b) an unknown preference, where the agent has to automatically infer $\boldsymbol{\omega}$. Efficiently aligning $\Pi_{\mathcal{L}}(\boldsymbol{\omega})$ with the preferred optimal policy is non-trivial since the CCS can be very large. In both cases, the agent is evaluated on how well it can adapt to tasks with unseen preferences.

## 2.1 Related Work

**Multi-Objective RL** Existing MORL algorithms can be roughly divided into two main categories [14, 15, 2]: *single-policy* methods and *multiple-policy* methods. Single-policy methods aim to find the optimal policy for a *given* preference among the objectives [16, 17]. These methods explore different forms of preference functions, including non-linear ones such as the minimum over all objectives or the number of objectives that exceed a certain threshold. However, single-policy methods do not work when preferences are unknown.

Multi-policy approaches learn a *set* of policies to obtain the approximate Pareto frontier of optimal solutions. The most common strategy is to perform multiple runs of a single-policy method over different preferences [7, 18]. Policy-based RL algorithms [19, 20] simultaneously learn the optimal manifold over a set of preferences. Several value-based reinforcement learning algorithms employ an extended version of the Bellman equation and maintain the convex hull of the discrete Pareto frontier [8, 21, 22]. Multi-objective fitted Q-iteration (MOFQI) [23, 24] encapsulates preferences as input to a Q-function approximator and uses expanded historical trajectories to learn multiple policies. This allows the agent to construct the optimal policy for any given preference during testing. However, these methods explicitly maintain sets of policies, and hence are difficult to scale up to high-dimensional preference spaces. Furthermore, these methods are designed to work during the learning phase but cannot be easily adapted to new preferences at test time.

**Scalarized Q-Learning.** Recent work has proposed the scalarized Q-learning algorithm [9] which uses a vector value function but performs updates after computing the inner product of the value function with a preference vector. This method uses an outer loop to perform a search over preferences,

while the inner loop performs the scalarized updates. Recently, Abels et al. [13] extended this to use a single neural network to represent value functions over the entire space of preferences. However, scalarized updates are not sample efficient and lead to sub-optimal MORL policies – our approach uses a global optimality filter to perform envelope Q-function updates, leading to faster and better learning (as we demonstrate in Figure 2(c) and Section 4).

Three key contributions distinguish our work from Abels et al. [13]: (1) At algorithmic level, our envelope Q-learning algorithm utilizes the convex envelope of the solution frontier to update parameters of the policy network, which allows our method to quickly align one preference with optimal rewards and trajectories that may have been explored under other preferences. (2) At theoretical level, we introduce a theoretical framework for designing and analyzing value-based MORL algorithms, and convergence proofs for our envelope Q-learning algorithm. (3) At empirical level, we provide new evaluation metrics and benchmark environments for MORL and apply our algorithm to a wider variety of domains including two complex larger scale domains – task-oriented dialog and supermario. Our FTN domain is a scaled up, more complex version of Minecart in [13].

**Policy Adaptation.** Our policy adaptation scheme is related to prior work in preference elicitation [25, 26, 27] or inverse reinforcement learning [28, 29]. Inverse RL (IRL) aims to learn a scalar reward function from expert demonstrations, or directly imitate the expert's policy without intermediate steps for solving a scalar reward function [30]. Chajewska et al. [31] proposed a Bayesian version to learn the utility function. IRL is effective when the hidden preference is fixed and expert demonstrations are available. In contrast, we require policy adaptation across various different preferences and do not use any demonstrations.

# 3  Multi-objective RL with Envelope Value Updates

In this section, we propose a new algorithm for multi-objective RL called *envelope Q-learning*. Our key idea is to use vectorized value functions and perform *envelope updates*, which utilize the convex envelope of the solution frontier to update parameters. This is in contrast to approaches like *scalarized Q-Learning*, which perform value function updates using only a single preference at a time. Since we learn a set of policies simultaneously over multiple preferences, and our concept of optimality is defined on vectorized rewards, existing convergence results from single-objective RL no longer hold. Hence, we first provide a theoretical analysis of our proposed update scheme below followed by a sketch of the resulting algorithm.

**Bellman operators.** The standard Q-Learning [32] algorithm for single-objective RL utilizes the Bellman optimality operator $T$:

$$(TQ)(s,a) := r(s,a) + \gamma \mathbb{E}_{s' \sim \mathcal{P}(\cdot|s,a)}(HQ)(s'). \tag{1}$$

where the operator $H$ is defined by $(HQ)(s') := \sup_{a' \in \mathcal{A}} Q(s',a')$ is an optimality filter over the Q-values for the next state $s'$.

We extend this to the MORL case by considering a value space $\mathcal{Q} \subseteq (\Omega \to \mathbb{R}^m)^{\mathcal{S} \times \mathcal{A}}$, containing all bounded functions $\boldsymbol{Q}(s,a,\omega)$ – estimates of expected total rewards under $m$-dimensional preference ($\boldsymbol{\omega}$) vectors. We can define a corresponding value metric $d$ as:

$$d(\boldsymbol{Q}, \boldsymbol{Q}') := \sup_{\substack{s \in \mathcal{S}, a \in \mathcal{A} \\ \boldsymbol{\omega} \in \Omega}} |\boldsymbol{\omega}^{\mathsf{T}}(\boldsymbol{Q}(s,a,\boldsymbol{\omega}) - \boldsymbol{Q}'(s,a,\boldsymbol{\omega}))|. \tag{2}$$

Since the identity of indiscernibles [33] does not hold, we note that $d$ forms a complete pseudo-metric space, and refer to $\boldsymbol{Q}$ as a *Multi-Objective Q-value (MOQ) function*. Given a policy $\pi$ and sampled trajectories $\tau$, we first define a multi-objective evaluation operator $\mathcal{T}_\pi$ as:

$$(\mathcal{T}_\pi \boldsymbol{Q})(s,a,\boldsymbol{\omega}) := \boldsymbol{r}(s,a) + \gamma \mathbb{E}_{\tau \sim (\mathcal{P}, \pi)} \boldsymbol{Q}(s',a',\boldsymbol{\omega}). \tag{3}$$

We then define an optimality filter $\mathcal{H}$ for the MOQ function as $(\mathcal{H}\boldsymbol{Q})(s,\boldsymbol{\omega}) := \arg_Q \sup_{a \in \mathcal{A}, \boldsymbol{\omega}' \in \Omega} \boldsymbol{\omega}^{\mathsf{T}} \boldsymbol{Q}(s,a,\boldsymbol{\omega}')$, where the $\arg_Q$ takes the multi-objective value corresponding to the supremum (i.e., $\boldsymbol{Q}(s,a,\boldsymbol{\omega}')$ such that $(a,\boldsymbol{\omega}') \in \arg \sup_{a \in \mathcal{A}, \boldsymbol{\omega}' \in \Omega} \boldsymbol{\omega}^{\mathsf{T}} \boldsymbol{Q}(s,a,\boldsymbol{\omega}'))$. The return of $\arg_Q$ depends on which $\boldsymbol{\omega}$ is chosen for scalarization, and we keep $\arg_Q$ for simplicity. This can be thought of as generalized version of the single-objective optimality filter in Eq. 1. Intuitively, $\mathcal{H}$ solves the convex envelope (hence the name envelope Q-learning) of the current solution frontier to

produce the $\boldsymbol{Q}$ that optimizes utility given state $s$ and preference $\boldsymbol{\omega}$. This allows for more optimistic Q-updates compared to using just the standard Bellman filter ($H$) that optimizes over actions only – this is the update used by scalarized Q-learning [13]. We can then define a *multi-objective optimality operator* $\mathcal{T}$ as:

$$(\mathcal{T}\boldsymbol{Q})(s, a, \boldsymbol{\omega}) := \boldsymbol{r}(s, a) + \gamma \mathbb{E}_{s' \sim \mathcal{P}(\cdot|s,a)}(\mathcal{H}\boldsymbol{Q})(s', \boldsymbol{\omega}). \tag{4}$$

The following theorems demonstrate the feasibility of using our optimality operator for multi-objective RL. Proofs for all the theorems are provided in the supplementary material.

**Theorem 1** (Fixed Point of Envelope Optimality Operator). *Let $\boldsymbol{Q}^* \in \mathcal{Q}$ be the preferred optimal value function in the value space, such that*

$$\boldsymbol{Q}^*(s, a, \boldsymbol{\omega}) = \arg_Q \sup_{\pi \in \Pi} \boldsymbol{\omega}^\intercal \mathbb{E}_{\substack{\tau \sim (\mathcal{P}, \pi) \\ |s_0=s, a_0=a}} \left[ \sum_{t=0}^{\infty} \gamma^t \boldsymbol{r}(s_t, a_t) \right], \tag{5}$$

*where the $\arg_Q$ takes the multi-objective value corresponding to the supremum. Then, $\boldsymbol{Q}^* = \mathcal{T}\boldsymbol{Q}^*$.*

Theorem 1 tells us the preferred optimal value function is a fixed-point of $\mathcal{T}$ in the value space.

**Theorem 2** (Envelope Optimality Operator is a Contraction). *Let $\boldsymbol{Q}, \boldsymbol{Q}'$ be any two multi-objective Q-value functions in the value space $\mathcal{Q}$ as defined above. Then, the Lipschitz condition $d(\mathcal{T}\boldsymbol{Q}, \mathcal{T}\boldsymbol{Q}') \leq \gamma d(\boldsymbol{Q}, \boldsymbol{Q}')$ holds, where $\gamma \in [0, 1)$ is the discount factor of the underlying MOMDP $\mathcal{M}$.*

Finally, we provide a generalized version of Banach's Fixed-Point Theorem in the pseudo-metric space.

**Theorem 3** (Multi-Objective Banach Fixed-Point Theorem). *If $\mathcal{T}$ is a contraction mapping with Lipschitz coefficient $\gamma$ on the complete pseudo-metric space $\langle \mathcal{Q}, d \rangle$, and $\boldsymbol{Q}^*$ is defined as in Theorem 1, then $\lim_{n \to \infty} d(\mathcal{T}^n \boldsymbol{Q}, \boldsymbol{Q}^*) = 0$ for any $\boldsymbol{Q} \in \mathcal{Q}$.*

Theorems 1-3 guarantee that iteratively applying optimality operator $\mathcal{T}$ on any MOQ-value function will terminate with a function $\boldsymbol{Q}$ that is equivalent to $\boldsymbol{Q}^*$ under the measurement of pseudo-metric $d$. These $\boldsymbol{Q}$s are as good as $\boldsymbol{Q}^*$ since they all have the same utilities for each $\boldsymbol{\omega}$, and will only differ when the utility corresponds to a recess in the frontier (see Figure 2(c) for an example, at the recess, either D or F is optimal).

Maintaining the envelope $\sup_{\boldsymbol{\omega}'} \boldsymbol{\omega}^\intercal Q(\cdot, \cdot, \boldsymbol{\omega}')$ allows our method to quickly align one preference with optimal rewards and trajectories that may have been explored under other preferences, while scalarized updates that optimizes the scalar utility cannot use the information of $\max_a Q(s, a, \boldsymbol{\omega}')$ to update the optimal solution aligned with a different $\boldsymbol{\omega}$. As illustrated in Figure 2 (c), assuming we have found two optimal solutions D and F in the CCS, misaligned with preferences $\boldsymbol{\omega}_2$ and $\boldsymbol{\omega}_1$. The scalarized update cannot use the information of $\max_a Q(s, a, \boldsymbol{\omega}_1)$ (corresponding to F) to update the optimal solution aligned with $\boldsymbol{\omega}_2$ or vice versa. It only searches along $\boldsymbol{\omega}_1$ direction leading to non-optimal L, even if solution D has been seen under $\boldsymbol{\omega}_2$. Hence, the envelope updates can have better sample efficiency in theory, as is also seen from the empirical results.

**Learning Algorithm.** Using the above theorems, we provide a sample-efficient learning algorithm for multi-objective RL (Algorithm 1). Since our goal is to induce a single model that can adapt to the entire space of $\Omega$, we use one parameterized function to represent $\mathcal{Q} \subseteq (\Omega \to \mathbb{R}^m)^{\mathcal{S} \times \mathcal{A}}$. We achieve this by using a deep neural network with $s, \boldsymbol{\omega}$ as input and $|\mathcal{A}| \times m$ Q-values as output. We then minimize the following loss function at each step $k$:[2]

$$L^\mathtt{A}(\theta) = \mathbb{E}_{s,a,\boldsymbol{\omega}} \left[ \| \boldsymbol{y} - \boldsymbol{Q}(s, a, \boldsymbol{\omega}; \theta) \|_2^2 \right], \tag{6}$$

where $\boldsymbol{y} = \mathbb{E}_{s'}[\boldsymbol{r} + \gamma \arg_Q \max_{a,\boldsymbol{\omega}'} \boldsymbol{\omega}^\intercal \boldsymbol{Q}(s', a, \boldsymbol{\omega}'; \theta_k)]$, which empirically can be estimated by sampling transition $(s, a, s', r)$ from a replay buffer.

Optimizing $L^\mathtt{A}$ directly is challenging in practice because the optimal frontier contains a large number of discrete solutions, which makes the landscape of loss function considerably non-smooth. To address this, we use an auxiliary loss function $L^\mathtt{B}$:

$$L^\mathtt{B}(\theta) = \mathbb{E}_{s,a,\boldsymbol{\omega}}[|\boldsymbol{\omega}^\intercal \boldsymbol{y} - \boldsymbol{\omega}^\intercal \boldsymbol{Q}(s, a, \boldsymbol{\omega}; \theta)|]. \tag{7}$$

Combined, our final loss function is $L(\theta) = (1 - \lambda) \cdot L^{\mathtt{A}}(\theta) + \lambda \cdot L^{\mathtt{B}}(\theta)$, where $\lambda$ is a weight to trade off between losses $L^{\mathtt{A}}$ and $L^{\mathtt{B}}_k$. We slowly increase the value of $\lambda$ from 0 to 1, to shift our loss function from $L^{\mathtt{A}}$ to $L^{\mathtt{B}}$. This method, known as *homotopy optimization* [12], is effective since for each update step, it uses the optimization result from the previous step as the initial guess. $L^{\mathtt{A}}$ first ensures the prediction of $\boldsymbol{Q}$ is close to any real expected total reward, although it may not be optimal. $L^{\mathtt{B}}$ provides an auxiliary pull along the direction with better utility.

The loss function above has an expectation over $\boldsymbol{\omega}$ – this entails sampling random preferences in the algorithm. However, since the $\boldsymbol{\omega}$s are decoupled from the transitions, we can increase sample efficiency by using a scheme similar to Hindsight Experience Replay [11]. Furthermore, computing the optimality filter $\mathcal{H}$ over the entire $\mathcal{Q}$ is infeasible; instead we approximate this by applying $\mathcal{H}$ over a mini-batch of transitions before performing parameter updates. Further details on our model architectures and implementation details are available in the supplementary material (Section A.2.3).

---

**Algorithm 1:** Envelope MOQ-Learning

**Input:** a preference sampling distribution $\mathcal{D}_\omega$, path $p_\lambda$ for the balance weight $\lambda$ increasing from 0 to 1.

Initialize replay buffer $\mathcal{D}_\tau$, network $\boldsymbol{Q}_\theta$, and $\lambda = 0$.

**for** *episode* = $1, \ldots, M$ **do**

  Sample a linear preference $\boldsymbol{\omega} \sim \mathcal{D}_\omega$.

  **for** $t = 0, \ldots, N$ **do**

    Observe state $s_t$.

    Sample an action $\epsilon$-greedily:

$$a_t = \begin{cases} \text{random action in } \mathcal{A}, & \text{w.p. } \epsilon; \\ \max_{a \in \mathcal{A}} \boldsymbol{\omega}^\mathsf{T} \boldsymbol{Q}(s_t, a, \boldsymbol{\omega}; \theta), & \text{w.p } 1 - \epsilon. \end{cases}$$

    Receive a vectorized reward $\boldsymbol{r}_t$ and observe $s_{t+1}$.

    Store transition $(s_t, a_t, \boldsymbol{r}_t, s_{t+1})$ in $\mathcal{D}_\tau$.

    **if** *update* **then**

      Sample $N_\tau$ transitions $(s_j, a_j, \boldsymbol{r}_j, s_{j+1}) \sim \mathcal{D}_\tau$.

      Sample $N_\omega$ preferences $W = \{\boldsymbol{\omega}_i \sim \mathcal{D}_\omega\}$.

      Compute $y_{ij} = (\mathcal{T}\boldsymbol{Q})_{ij} =$

$$\begin{cases} \boldsymbol{r}_j, & \text{for terminal } s_{j+1}; \\ \boldsymbol{r}_j + \gamma \arg_Q \max\limits_{\substack{a \in \mathcal{A}, \\ \boldsymbol{\omega}' \in W}} \boldsymbol{\omega}_i^\mathsf{T} \boldsymbol{Q}(s_{j+1}, a, \boldsymbol{\omega}'; \theta), & \text{o.w.} \end{cases}$$

      for all $1 \le i \le N_\omega$ and $1 \le j \le N_\tau$.

      Update $Q_\theta$ by descending its stochastic gradient according to equations 6 and 7:

$$\nabla_\theta L(\theta) = (1 - \lambda) \cdot \nabla_\theta L^{\mathtt{A}}(\theta) + \lambda \cdot \nabla_\theta L^{\mathtt{B}}(\theta).$$

    Increase $\lambda$ along the path $p_\lambda$.

---

**Policy adaptation.** Once we obtain a policy model $\Pi_{\mathcal{L}}(\boldsymbol{\omega})$ from the learning phase, the agent can adapt to any provided preference by simply feeding the $\boldsymbol{\omega}$ into the network. While this is a straightforward scenario, we also consider a more challenging test where only scalar rewards are available and the agent has to uncover a hidden preference $\boldsymbol{\omega}$ while adapting to the new task. For this case, we assume preferences are drawn from a truncated multivariable Gaussian distribution $\mathcal{D}_\omega^m(\mu_1, \ldots, \mu_m; \sigma)$ on an $(m-1)$-simplex, where nonnegative parameters $\mu_1, \ldots, \mu_m$ are the means with $\mu_1 + \cdots + \mu_m = 1$, and $\sigma$ is a fixed standard deviation for all dimensions. Our goal is then to infer the parameters of this Gaussian distribution, for which we perform a combination of policy gradient (e.g., REINFORCE [35]) and stochastic search while keeping the policy model fixed. We determine the best preference parameters that maximize the expected return in the target task:

$$\arg\max_{\mu_1, \ldots, \mu_m} \mathbb{E}_{\boldsymbol{\omega} \sim \mathcal{D}_\omega^m} \left[ \mathbb{E}_{\tau \sim (\mathcal{P}, \Pi_{\mathcal{L}}(\boldsymbol{\omega}))} \left[ \sum_{t=0}^{\infty} \gamma^t r_t(s_t, a_t) \right] \right]. \tag{8}$$

## 4 Experiments

**Evaluation Metrics.** Three metrics are to evaluate the empirical performance on test tasks:

a) *Coverage Ratio (CR)*. The first metric is *coverage ratio* (CR), which evaluates the agent's ability to recover optimal solutions in the convex coverage set (CCS). If $\mathcal{F} \subseteq \mathbb{R}^m$ is the set of solutions found by the agent (via sampled trajectories), we define $\mathcal{F} \cap_\epsilon \mathtt{CCS} := \{x \in \mathcal{F} \mid \exists y \in \mathtt{CCS} \text{ s.t. } \|x - y\|_1 / \|y\|_1 \le \epsilon\}$ as the intersection between these sets with a tolerance of $\epsilon$. The CR is then defined as:

$$\mathtt{CR}_{\mathtt{F1}}(\mathcal{F}) := 2 \cdot \frac{\mathtt{precision} \cdot \mathtt{recall}}{\mathtt{precision} + \mathtt{recall}}, \tag{9}$$

where the $\mathtt{precision} = |\mathcal{F} \cap_\epsilon \mathtt{CCS}| / |\mathcal{F}|$, indicating the fraction of optimal solutions among the retrieved solutions, and the $\mathtt{recall} = |\mathcal{F} \cap_\epsilon \mathtt{CCS}| / |\mathtt{CCS}|$, indicating the fraction of optimal instances that have been retrieved over the total amount of optimal solutions (see Figure 3(a)).

b) *Adaptation Error (AE)*. Our second metric compares the retrieved control frontier with the optimal one, when an agent is provided with a specific preference $\boldsymbol{\omega}$ during the adaptation phase:

$$\mathrm{AE}(\mathcal{C}) := \mathbb{E}_{\boldsymbol{\omega} \sim \mathcal{D}_{\boldsymbol{\omega}}}[|\mathcal{C}(\boldsymbol{\omega}) - \mathcal{C}_{\mathrm{opt}}(\boldsymbol{\omega})|/\mathcal{C}_{\mathrm{opt}}(\boldsymbol{\omega})], \tag{10}$$

which is the expected relative error between optimal control frontier $\mathcal{C}_{\mathrm{opt}} : \Omega \to \mathbb{R}$ with $\boldsymbol{\omega} \mapsto \max_{\hat{\boldsymbol{r}} \in \mathrm{CCS}} \boldsymbol{\omega}^{\mathsf{T}} \hat{\boldsymbol{r}}$ and the agent's control frontier $\mathcal{C}_{\pi_{\boldsymbol{\omega}}} = \boldsymbol{\omega}^{\mathsf{T}} \hat{\boldsymbol{r}}_{\pi_{\boldsymbol{\omega}}}$.

c) *Average Utility (UT)*. This measures the average utility obtained by the trained agent on randomly sampled preferences and is a useful proxy to AE when we don't have access to the optimal policy.

**Domains.** We evaluate on four different domains (complete details in supplementary material):

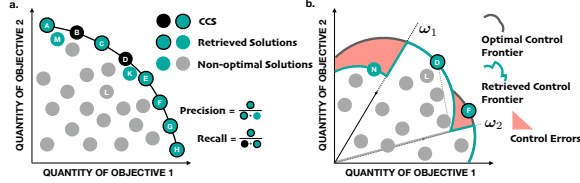

**Figure 3:** Illustration of evaluation metrics for MORL. (a.) Coverage ratio (CR) measures an agent's ability to find all the potential optimal solutions in the convex coverage set of Pareto frontier. Dots with black boundary are solutions in CCS, dots without black boundary are non-optimal returns, and dots in green are solutions retrieved by an MORL algorithm. CR is the F1 based on the precision and recall calculation. (b.) Adaptation error (AE) measures an agent's ability of policy adaptation to real-time specified preferences. The gray curve indicates the theoretical limit of the best cumulative utilities under all preference, and the green curve indicates the cumulative utilities of an MORL algorithm. AE is the average gap between these two curves over all preferences.

1. **Deep Sea Treasure (DST)** A classic MORL benchmark [14] in which an agent controls a submarine searching for treasures in a $10 \times 11$-grid world while trading off `time-cost` and `treasure-value`. The grid world contains 10 treasures of different values. Their values increase as their distances from the starting point $s_0 = (0, 0)$ increase. We ensure the Pareto frontier of this environment to be convex.

2. **Fruit Tree Navigation (FTN)** A full binary tree of depth $d$ with randomly assigned vectorial reward $\boldsymbol{r} \in \mathbb{R}^6$ on the leaf nodes. These rewards encode the amounts of six different components of nutrition of the *fruits* on the tree: {`Protein`, `Carbs`, `Fats`, `Vitamins`, `Minerals`, `Water`}. For every leaf node, $\exists \boldsymbol{\omega}$ for which its reward is optimal, thus all leaves lie on the CCS. The goal of our MORL agent is to find a path from the root to a leaf node that maximizes utility for a given preference, choosing between left or right subtrees at every non-terminal node.

3. **Task-Oriented Dialog Policy Learning (Dialog)** A modified task-oriented dialog system in the restaurant reservation domain based on PyDial [36]. We consider the task success rate and the dialog brevity (measured by number of turns) as two competing objectives of this domain.

4. **Multi-Objective SuperMario Game (SuperMario)** A multi-objective version of the popular video game Super Mario Bros. We modify the open-source environment from OpenAI gym [37] to provide vectorized rewards encoding five different objectives: `x-pos`: value corresponding to the difference in Mario's horizontal position between current and last time point, `time`: a small negative time penalty, `deaths`: a large negative penalty given each time Mario dies , `coin`: rewards for collecting coins, and `enemy`: rewards for eliminating an enemy.

**Baselines.** We compare our envelope MORL algorithm with classic and state-of-the-art baselines:

1. **MOFQI** [24]: Multi-objective fitted Q-iteration where the Q-approximator is a large linear model.

2. **CN+OLS** [13]: Conditional neural network with Optimistic Linear Support (OLS) method as the outer loop for selecting $\boldsymbol{\omega}$. This method is first proposed in [9] with multiple neural networks, and we employ an improved version using single conditional neural network [13].

3. **Scalarized** [13]: The state-of-the-art algorithm uses scalarized Q-update with double Q-learning, prioritized and hindsight experience replay, which is equivalent to CN+DER proposed in [13].

**Main Results.** Table 1 shows the performance comparison of different MORL algorithms in four domains. We elaborate training and test details for each domain in supplementary material. In DST and FTN we compare CR and AE as defined in section 4. In the task-oriented dialog policy learning task, we compare the average utility (Avg. UT) for 5,000 test dialogues with uniformly sampled user preferences on success and brevity. In the SuperMario game, the Avg. UT is over 500 test episodes

| Method | DST | | FTN ($d = 6$) | | Dialog[2] | SuperMario[2] |
|---|---|---|---|---|---|---|
| | CR ↑ | AE ↓ | CR ↑ | AE ↓ | Avg.UT ↑ | Avg.UT ↑ |
| MOFQI | $0.639 \pm 0.421$ | $139.6 \pm 25.98$ | $0.197 \pm 0.000$ | $0.176 \pm 0.001$ | $2.17 \pm 0.21$ | – |
| CN+OLS | $0.751 \pm 0.163$ | $34.63 \pm 1.396$ | – | – | $2.53 \pm 0.22$ | – |
| Scalarized | $0.989 \pm 0.024$ | $0.165 \pm 0.096$ | $0.914 \pm 0.044$ | $0.016 \pm 0.005$ | $2.38 \pm 0.22$ | $162.7 \pm 77.66$ |
| Envelope (ours)[1] | $\mathbf{0.994 \pm 0.001}$ | $\mathbf{0.152 \pm 0.006}$ | $\mathbf{0.987 \pm 0.021}$ | $\mathbf{0.006 \pm 0.001}$ | $\mathbf{2.65 \pm 0.22}$ | $\mathbf{321.2 \pm 146.9}$ |

**Table 1:** Comparison of different MORL algorithms in learning and adaptation phases across four experimental domains. ↑ indicates higher is better, and ↓ indicates lower is better for the scores. Each data point indicates the mean and standard deviation over 5 independent training and test runs. [1]Using the unpaired t-test, we obtain significance scores of $p < 0.05$ vs MOFQI on all domains, $p < 0.01$ vs CN+OLS on DST and $p < 0.05$ vs Scalarized on FTN, Dialog and SuperMario. [2]Additional results are in the supplementary material C.4 and C.5.

with uniformly sampled preferences. The envelope algorithm steadily achieves the best performance in terms of both learning and adaptation among all the MORL methods in all four domains.

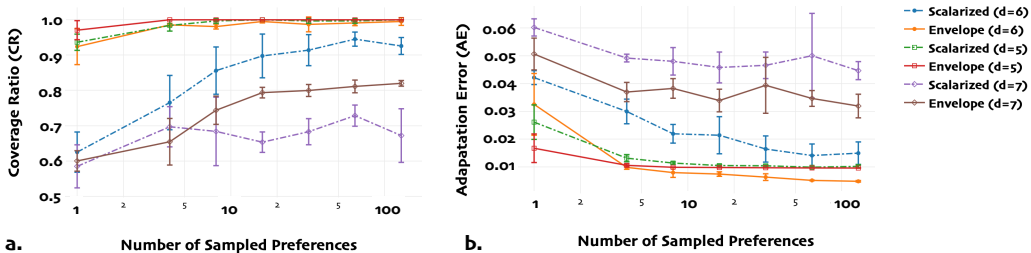

**Figure 4:** Coverage Ratio (CR) and Adaptation Error (AE) comparison of the scalarized algorithm [13] and our envelope deep MORL algorithm over 5000 episodes of FTN tasks of depths $d = 5, 6, 7$. Higher CR indicates better coverage of optimal policies, lower AE indicates better adaptation. The error bars are standard deviations of CR and AE estimated from 5 independent runs under each configuration.

**Scalability.** There are three aspects of the scalability of a MORL algorithm: the ability to deal with (1) large state space, (2) many objectives, and (3) large optimal policy set. Unlike other neural network-based methods, MOFQI cannot deal with the large state space, e.g., the video frames in SuperMario Game. The CN+OLS baseline requires solving all the intersection points of a set of hyper-planes thus is computationally intractable in domains with $m > 3$ objectives, such as FTN and SuperMario. We denote these entries as "–" in Table 1. Both scalarized and envelope methods can be applied to cases having large state space and reasonably many objectives. However, the size of optimal policy set may affect the performance of these algorithms. Figure 4 shows CR and AE results in three FTN environments with $d = 5$ (with 32 solutions), $d = 6$ (with 64 solutions), and $d = 7$ (with 128 solutions). We observe that both scalarized and envelope algorithms are close to optimal when $d = 5$ but both CR and AE values are worse for $d = 7$. However, the envelope version is more stable and outperforms the scalarized MORL algorithm in all three cases. These results point to the robustness and scalability of our algorithms.

**Sample Efficiency.** To compare sample efficiency during the learning phase, we train both our scalarized and envelope deep MORL on the FTN task with different depths for 5,000 episodes. We compute coverage ratio (CR) over 2,000 episodes and adaptation error (AE) over 5,000 episodes. Figure 4 shows plots for the metrics computed over a varying number of sampled preferences $N_\omega$ (more details can be found in the supplementary material). Each point on the curve is averaged over 5 experiments. We observe that the envelope MORL algorithm consistently has a better CR and AE scores than the scalarized version, with smaller variances. As $N_\omega$ increases, CR increases and AE decreases, which shows better use of historical interactions for both algorithms when $N_\omega$ is larger. And to achieve the same level AE the envelope algorithm requires smaller $N_\omega$ than the scalarized algorithm. This reinforces our theoretical analysis that the envelope MORL algorithm has better sample efficiency than the scalarized version.

**Policy Adaptation.** We show how the MORL agents respond to user preference during the adaptation phase in the dialog policy learning task, where the agent must trade off between the dialog success rate and the conversation brevity. Figure 5 shows the success rate (SR) curves as we vary the

|  | Protein | Carbs | Fats | Vitamins | Minerals | Water |
|---|---|---|---|---|---|---|
| v1 | 0.9639 | 0. | 0.0361 | 0. | 0. | 0. |
| v2 | 0. | 0.9067 | 0. | 0. | 0. | 0.0933 |
| v3 | 0.0539 | 0. | 0.9461 | 0. | 0. | 0. |
| v4 | 0.1366 | 0.0459 | 0. | 0.7503 | 0.0671 | 0. |
| v5 | 0. | 0.0148 | 0.0291 | 0.0428 | 0.7503 | 0.1629 |
| v6 | 0. | 0.0505 | 0. | 0. | 0. | 0.9495 |

**Table 2:** Inferred preferences of the envelope MOQ-learning algorithm on different FTN ($d = 6$) tasks (v1 to v6) after only 15 episodes interaction. The underlying preferences are all ones on the diagonal of the table and zeros for the off-diagonal.

|  | x-pos | time | life | coin | enemy |
|---|---|---|---|---|---|
| g1 | 0.5288 | 0.1770 | 0.1500 | 0.0470 | 0.0972 |
| g2 | 0.1985 | 0.2237 | 0.2485 | 0.1422 | 0.1868 |
| g3 | 0.2196 | 0.1296 | 0.3541 | 0.1792 | 0.1175 |
| g4 | 0.0211 | 0.2404 | 0.0211 | 0.6960 | 0.0211 |
| g5 | 0.0715 | 0.1038 | 0.2069 | 0.3922 | 0.2253 |

**Table 3:** Inferred preferences of the envelope multi-objective A3C algorithm in different Mario Game variants (g1 to g5) with 100 episodes. The underlying preferences are all ones on the diagonal of the table and zeros for the off-diagonal.

weight of the preference on task completion success. The success rates of both MORL algorithms increase as the user's weight on success increases, while those of the single-objective algorithms do not change. This shows that our envelope MORL agent can adapt gracefully to the user's preference. Furthermore, our envelope deep MORL algorithm outperforms other algorithms whenever success is relatively more important to the user (weight $> 0.5$).

**Revealing underlying preferences.** Finally, we test the ability of our agent to infer and adapt to unknown preferences on FTN and SuperMario. During the learning phase, the agent does not know the underlying preference, and hence learns a multi-objective policy. During the adaptation phase, our agent performs recovers underlying preferences (as described in Section 3) to uncover the underlying preference that maximizes utility. Table 2 shows the learned preferences for 6 different FTN tasks (v1 to v6) with unknown one-hot preferences $[1, 0, 0, 0, 0, 0]$ to $[0, 0, 0, 0, 0, 1]$, respectively, meaning the agent should only care about one elementary nutrition. These were learned in a few-shot adaption setting, using just 15 episodes. For the SuperMario Game, we implement an A3C [38] variant of our envelope

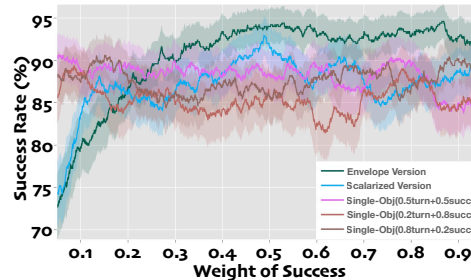

**Figure 5:** The success-weight curves of task-oriented dialog. Each data point is a moving average of closest around 500 dialogues in the interval of around $\pm$ 0.05 weight of success. The light shadow indicates the standard deviations of 5 independent runs under each configuration.

MORL agent (see supplementary material for details). Table 3 shows the learned preferences for 5 different tasks (g1 to g5) with unknown one-hot preferences using just 100 episodes.

We observe that the learned preferences are concentrated on the diagonal, indicating good alignment with the actual underlying preferences. For example, in the SuperMario game variant g4, the envelope MORL agent finds the preference with the highest weight (0.6960) on the coin objective can best describe the goal of g4, which is to collect as many coins as possible. We also tested policy adaptation on the original Mario game using game scores for the scalar rewards. We find that the agent learns preference weights of 0.37 for x-pos and 0.23 for time, which seems consistent with a common strategy that humans employ – simply move Mario towards the flag as quickly as possible.

## 5   Conclusion

We have introduced a new algorithm for multi-objective reinforcement learning (MORL) with linear preferences, with the goal of enabling few-shot adaptation of autonomous agents to new scenarios. Specifically, we propose a multi-objective version of the Bellman optimality operator, and utilize it to learn a single parametric representation for all optimal policies over the space of preferences. We provide convergence proofs for our multi-objective algorithm and also demonstrate how to use our model to adapt and elicit an unknown preference on a new task. Our experiments across four different domains demonstrate that our algorithms exhibit effective generalization and policy adaptation.

## Acknowledgements

The authors would like to thank Yexiang Xue at Purdue University, Carla Gomes at Cornell University for helpful discussions on multi-objective optimization, Lu Chen, Kai Yu at Shanghai Jiao Tong University for discussing dialogue applications, Haoran Cai at MIT for helping running a part of synthetic experiments, and anonymous reviewers for constructive suggestions.

## Footnotes

[1]Code is available at https://github.com/RunzheYang/MORL

[2]We use double Q learning with target Q networks following Mnih et al. [34]

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
