[Supplementary Material]

# Supplementary Material for Generalized Algorithm for Multi-Objective RL and Policy Adaptation

## A Theoretical Framework for Value-Based MORL Algorithms

In this section, we introduce a theoretical framework for analyzing and designing the value-based multi-objective reinforcement learning algorithms. This framework is based on the well-known Banach's Fixed-Point Theorem, which guarantees the existence and uniqueness of fixed-point of a *contraction* on a *complete metric space*. Therefore, generalizing this theorem a bit, we can imagine all value functions of reinforcement learning are in some metric space, and finding the optimal value or policy is to find the fixed point of a certain contraction on that space. We first recall the following concepts.

### A.1 General Framework for Value-Based Reinforcement Learning

**Definition 1** (Contraction). *Let $(X, d)$ be a metric space. We say that $\mathcal{T}$ is a* contraction*, if there is a real number $\gamma \in [0, 1)$ such that*

$$d(\mathcal{T}(x), \mathcal{T}(x')) \leq \gamma d(x, x') \tag{11}$$

*for all points $x, x' \in X$, where $\gamma$ is called a Lipschitz coefficient for the contraction $\mathcal{T}$.*

**Theorem 4** (Banach's Fixed-Point Theorem). *Let $(X, d)$ be a complete metric space and let $\mathcal{T} : X \to X$ be a contraction. Then there exists a **unique** fixed point $x^* \in X$ such that $\mathcal{T}(x^*) = x^*$. Moreover, if $x$ is any point in $X$ and $\mathcal{T}^n(x)$ is inductively defined by $\mathcal{T}^n(x) = \mathcal{T}(\mathcal{T}^{n-1}(x))$, then we have $\mathcal{T}^n(x) \to x^*$ as $n \to \infty$.*

The above introduced Banach fixed-point theorem is well-known. Readers may refer to the book [39] for more details. Practically, this provides us with an iterative method for converging to any desired solution in the large solution space, by repeatedly applying a properly designed contraction. For example, the foundation for standard value-based single-objective reinforcement learning is the use of Bellman's optimality equation [10]:

$$Q^*(s, a) = r(s, a) + \gamma \mathbb{E}_{s' \sim \mathcal{P}(\cdot|s,a)} \sup_{a' \in \mathcal{A}} Q^*(s', a'), \tag{12}$$

where $\gamma \in [0, 1)$ is the discount factor and the optimal Q-value function $Q^*(s, a)$ is the desired solution in the space $\mathcal{Q} \subseteq \mathbb{R}^{\mathcal{S} \times \mathcal{A}}$ consisting of all the bounded functions with $\ell_\infty$-distance metric

$$d(Q, Q') := \sup_{s \in \mathcal{S}, a \in \mathcal{A}} |Q(s, a) - Q'(s, a)|. \tag{13}$$

Since the all the functions in this space is bounded, it follows that with this $\ell_\infty$-distance metric, the space $(\mathcal{Q}, d)$ is complete. Besides, according to the equation (12), we can design an *Bellman optimality operator* $\mathcal{T}$ such that

$$(\mathcal{T}Q)(s, a) := r(s, a) + \gamma \mathbb{E}_{s' \sim \mathcal{P}(\cdot|s,a)} \sup_{a' \in \mathcal{A}} Q(s', a'), \tag{14}$$

which can be shown as a contraction on $(\mathcal{Q}, d)$. Many popular value-based reinforcement algorithms, such as *deep Q-learning* [34], can be seen as asynchronous iteration methods with approximately applied contraction.

We can verify that the Bellman optimality operator $\mathcal{T}$ indeed is a contraction on $(\mathcal{Q}, d)$, and the optimal value function $Q^*$ is a fixed point in $(\mathcal{Q}, d)$. Therefore, we can find the unique optimal Q-value function by applying the optimality operator iteratively many times on any initial Q-value function. Similarly, we can also define *Bellman evaluation operator* $\mathcal{T}_\pi$ using the Bellman expectation equation $(\mathcal{T}_\pi Q)(s, a) := r(s, a) + \gamma \mathbb{E}_{\tau \sim (\mathcal{P}, \pi)} Q(s', a')$, which is also a contraction.

Knowing that the optimality operator is a contraction is important. In practice, we use a minibatch to update previous Q-value function approximated by neural networks, not updating all states and actions. Thus the updated Q-value function is not a strict $\mathcal{T}Q$, but only close to $\mathcal{T}Q$ on some state and action pairs. We can still provide a theoretical guarantee that a minibatch iterative algorithm can still converge to a promising result, under certain extra assumptions.

**Definition 2** (Minibatch Iteration). *Consider the Q-value function $Q$ as a composition of $\{Q_{S,A}\}_{S \subseteq \mathcal{S}, A \subseteq \mathcal{A}}$ such that in each iteration,*

$$Q_{S,A}^{k+1}(s, a) := \begin{cases} \mathcal{T}Q_{S,A}^k(s, a), & \text{if } s \in S \text{ and } a \in A; \\ Q_{S,A}^k(s, a), & \text{otherwise.} \end{cases}$$

**Theorem 5** (Minibatch Convergence Theorem)**.** *Suppose each restricted Q-value function $Q_{S,A}$ can be update an arbitrary number of times, and there is a nest sequence of nonempty sets $\{\mathcal{Q}^k\}_{k \in \mathbb{Z}}$ with $\mathcal{Q}^{k+1} \subseteq \mathcal{Q}^k \subseteq \mathcal{Q}$, $k = 0, 1, \ldots$ such that if $\{Q^k\}_{k \in \mathbb{N}}$ is any sequence with $Q^k \in \mathcal{Q}^k$ for all $k \geq 0$, then $\{Q^k\}$ converges pointwisely to $Q^*$. Assume further the following:*

1. *Convergence Condition: We have*

$$\forall Q \in \mathcal{Q}^k, \mathcal{T}Q \in \mathcal{Q}^{k+1}; \tag{15}$$

2. *Box Condition: For all $k$, $\mathcal{Q}^k$ is a Cartesian product of the form*

$$\mathcal{Q}^k = \times_{s \in \mathcal{S}, a \in \mathcal{A}} \mathcal{Q}^k_{\{s\}, \{a\}}, \tag{16}$$

   *where $\mathcal{Q}^k_{S,A}$ is a set of bounded real-valued functions on states $S$ and actions $A$.*

*Then for every $Q^0 \in \mathcal{Q}^0$ the sequence $\{Q^k\}$ generated by the minibatch iteration algorithm converges to $Q^*$ [40].*

*Proof.* Showing the convergence of the algorithm is equivalent to showing that the iterations of elements from $\mathcal{Q}^k$ will get in to $\mathcal{Q}^{k+1}$ eventually, i.e., for each $k \geq 0$, there is a time $t_k$ such that $Q^t \in \mathcal{Q}^k$ for all $t \geq t_k$. We can prove it by mathematical induction.

When $k = 0$, the statement is true since $Q^0 \in \mathcal{Q}^0$. Assuming the statement is true for a given $k$, we will show there is a time $t_{k+1}$ with the required properties. For each $s \in \mathcal{S}, a \in \mathcal{A}$, let set $L_{s,a} = \{t : s^t = s, a^t = a\}$ record the time a minibatch update happens on the state $s$ and action $a$. Let $t(s, a)$ be the first element in $L_{s,a}$ such that $t(s, a) \geq t_k$. Then by the convergence condition, we have $\mathcal{T}Q^{t(s,a)} \in \mathcal{Q}^{k+1}$ for all $s \in \mathcal{S}$ and $a \in \mathcal{A}$. In the view of the box condition, it implies $Q^{t+1}(s, a) \in \mathcal{Q}^{k+1}_{s,a}$ for all $t \geq t(s, a)$ for any $s \in \mathcal{S}$ and $a \in \mathcal{A}$. Therefore, let $t_{k+1} = \max_{s,a} t(s, a) + 1$, using the box condition, we have $Q^t \in \mathcal{Q}^{k+1}$ for all $t \leq t_{k+1}$. By mathematical induction, the statement holds, and $\mathcal{Q}^k$ will shrink in to $\mathcal{Q}^{k+1}$ eventually. Hence, we have proved $\{Q^t\}$ converges to $Q^*$ finally.

$\square$

The above theorem indicates that minibatch update with experience replay will not affect the convergence of iteratively applying the optimality operator $\mathcal{T}$. This gives us the flexibility to design value-based algorithm's updating scheme. Besides, even though we use deep Q-network to approximate the Q-value function and update it by using gradient descent, this will not impair the magical function of optimality operator too much. This is because if after n-round neural network updates, we always have $\sup_{s \in \mathcal{S}, a \in \mathcal{A}} \left| Q^{t+n} - \mathcal{T}Q^t \right| \leq \epsilon$, where $n$ is an arbitrary bounded integer, by applying the triangle inequality we conclude that the final error $d(Q_{final}, Q^*) \leq \epsilon/(1 - \gamma)$ is bounded.

We summarize a general theoretical framework for value-based RL algorithm design, which consists of five elements:

- **Value Space $\mathcal{X}$**: A common choice of $\mathcal{X}$ is the space of value functions in $\mathbb{R}^{\mathcal{S}}$ or the space of Q-value functions in $\mathbb{R}^{\mathcal{S} \times \mathcal{A}}$. There are many other choices such as the space of ordered pair of $(\mathcal{V}, \mathcal{Q})$ (see 2.6.2 in book [41] for example), or the space of vectorized value functions as we show in the following sections.

- **Value Metric $d$**: It defines the "distance" between two points in the value space. Besides the basic four requirements, the metric $d$ should ensure a complete metric space $(\mathcal{X}, d)$ to validate the Banach's fixed-point theorem. A compatible selection of value metric will make the convergence analysis easier.

- **Evaluation Operator $\mathcal{T}_\pi$**: We have constructed a recursive expression of a certain value point in the value space associated with some policy, e.g., the Bellman expectation equation, to depict the value of a policy $v_\pi$ as a fixed point we desire. Carefully verify that the contraction property holds for $\mathcal{T}_\pi$.

- **Optimality Operator $\mathcal{T}$**: A recursive expression of the optimal point in the value space, e.g., the Bellman optimality equation. Note that when the metric $d$ is the supremum of the absolute value of the difference, and $\mathcal{T}_\pi$ is a contraction, we can prove the contraction property of $\mathcal{T}$ is always automatically satisfied.

- **Updating Scheme**: To make a reinforcement learning algorithm practical and scalable, we need to consider many factors in terms of updating scheme. For example: How do we approximate the value and policies? If it is an online algorithm, how do we trade off exploration and exploitation? All these details will significantly influence the performance of our algorithm on real-world tasks.

In summary, there are five essential components for analyzing and designing general value-base reinforcement learning algorithms: **(1) value space**, **(2) value metric**, **(3) evaluation operator**, **(4) optimality operator**, and **(5) updating scheme**. In fact, there is some work [42] developing distributional reinforcement learning in a way similar to this framework. We will discuss how to design these five elements of our framework to develop envelope multi-objective reinforcement learning algorithms in the next section.

## A.2 MORL with envelope updates

The deep MORL algorithm with scalarized update is capable of solving unknown linear preference scenarios of multi-objective reinforcement learning. However, there are several limitations of this algorithm restrict its applicability and performance in practice. Aiming at solving problems stated in Section 2, we design a new algorithm called envelope deep MORL algorithm. Following the value-based theoretical framework introduced in Section A.1, our key idea to upgrade the scalarized algorithm is to consider a different value space, where every Q-value function is a mapping to multi-objective solutions, not utilities, and therefore maintains the necessary information for prediction in the adaptation phase. Furthermore, we generalize the optimality filter $\mathcal{H}$ to use that information to boost up the alignment in the learning phase.

We consider a new value space $\mathcal{Q} \subseteq (\Omega \to \mathbb{R}^m)^{\mathcal{S} \times \mathcal{A}}$, containing all bounded functions $\boldsymbol{Q}(s, a, \boldsymbol{\omega})$ returning the estimates of preferred expected total rewards under preference $\boldsymbol{\omega}$, which are $m$-dimensional vectors. Besides, we employ a value metric $d$ defined by

$$d(\boldsymbol{Q}, \boldsymbol{Q}') := \sup_{\substack{s \in \mathcal{S}, a \in \mathcal{A} \\ \boldsymbol{\omega} \in \Omega}} |\boldsymbol{\omega}^\mathsf{T}(\boldsymbol{Q}(s, a, \boldsymbol{\omega}) - \boldsymbol{Q}'(s, a, \boldsymbol{\omega}))|. \tag{17}$$

Notice that this metric is a pseudo-metric, since the identity of indiscernibles does not hold for it. It is easy to show that metric space $(\mathcal{Q}, d)$ is complete.

We refer the Q-value functions in this $\mathcal{Q}$ *multi-objective value functions*. Similar to the scalarized one, we design an evaluation operator and an optimality operator for this envelope version algorithm. As for the updating scheme, we use hindsight experience replay and a homotopy optimization trick.

### A.2.1 Multi-Objective Bellman Optimality Operator

In this section, we give the evaluation operator and the optimality operator in the new envelope version value space $(\mathcal{Q}, d)$ as stated above. The evaluation operator now is even simpler than that of the scalarized version. Give a policy $\pi$, the evaluation operator is defined by

$$(\mathcal{T}_\pi \boldsymbol{Q})(s, a, \boldsymbol{\omega}) := \boldsymbol{r}(s, a) + \gamma \mathbb{E}_{\tau \sim (\mathcal{P}, \pi)} \boldsymbol{Q}(s', a', \boldsymbol{\omega}). \tag{18}$$

Since now the multi-objective Q-value function is also in a vector form, this evaluation operator is almost the same as that of the single-objective reinforcement learning. It can be easily verified as a contraction.

As for the envelope version of optimality operator, we employ a stronger optimality filter $\mathcal{H}$, defined by $(\mathcal{H}\boldsymbol{Q})(s, \boldsymbol{\omega}) := \arg_Q \sup_{a' \in \mathcal{A}, \boldsymbol{\omega}' \in \Omega} \boldsymbol{\omega}^\mathsf{T}\boldsymbol{Q}(s, a', \boldsymbol{\omega}')$, where the $\arg_Q$ takes the multi-objective value corresponding to the supremum, i.e., $\boldsymbol{Q}(s, a'', \boldsymbol{\omega}'')$ such that $(a'', \boldsymbol{\omega}'') \in \arg \sup_{a' \in \mathcal{A}, \boldsymbol{\omega}' \in \Omega} \boldsymbol{\omega}^\mathsf{T}\boldsymbol{Q}(s, a', \boldsymbol{\omega}')$. The return of $\arg_Q$ depends on scalarization weights $\boldsymbol{\omega}$, and we use $\arg_Q$ for simplicity of notation. This filter is solving the convex envelope of the current Pareto frontier, therefore we name this algorithm as "envelope" version. We can write the optimality operator $\mathcal{T}$ in terms of the optimal filter:

$$(\mathcal{T}\boldsymbol{Q})(s, a, \boldsymbol{\omega}) := \boldsymbol{r}(s, a) + \gamma \mathbb{E}_{s' \sim \mathcal{P}(\cdot|s,a)}(\mathcal{H}\boldsymbol{Q})(s', \boldsymbol{\omega}). \tag{19}$$

**Theorem 1** (Fixed Point of Evelope Optimality Operator for MORL)**.** *Use above definitions in the envelope version value space. Let $\boldsymbol{Q}^* \in \mathcal{Q}$ be the preferred optimal value function in the value space, such that*

$$\boldsymbol{Q}^*(s, a, \boldsymbol{\omega}) = \arg_Q \sup_{\pi \in \Pi} \boldsymbol{\omega}^\mathsf{T} \mathbb{E}_{\tau \sim (\mathcal{P}, \pi)|s_0=s, a_0=a} \left[ \sum_{t=0}^{\infty} \gamma^t \boldsymbol{r}(s_t, a_t) \right], \tag{20}$$

*where the $\arg_Q$ takes the multi-objective value corresponding to the supremum, then $\boldsymbol{Q}^* = \mathcal{T}\boldsymbol{Q}^*$ holds.*

*Proof.* First, we observe that $d(\boldsymbol{Q}^*, \mathcal{T}\boldsymbol{Q}^*) = \sup_{\substack{s \in \mathcal{S}, a \in \mathcal{A} \\ \boldsymbol{\omega} \in \Omega}} |\boldsymbol{\omega}^\mathsf{T}(\boldsymbol{Q}^*(s, a, \boldsymbol{\omega}) - \mathcal{T}\boldsymbol{Q}^*(s, a, \boldsymbol{\omega}))| = 0 \Leftrightarrow$
$\boldsymbol{\omega}^\mathsf{T}\mathcal{T}\boldsymbol{Q}^*(s, a, \boldsymbol{\omega}) = \boldsymbol{\omega}^\mathsf{T}\boldsymbol{Q}^*(s, a, \boldsymbol{\omega})$ for all $s, a, \boldsymbol{\omega}$. Then, by substituting the definition of $Q^*$ into $\mathcal{T}Q^*$,

$$
\begin{aligned}
\boldsymbol{\omega}^\mathsf{T}\mathcal{T}\boldsymbol{Q}^*(s, a, \boldsymbol{\omega}) &= \boldsymbol{\omega}^\mathsf{T}\boldsymbol{r}(s, a) + \gamma \cdot \boldsymbol{\omega}^\mathsf{T}\mathbb{E}_{s' \sim \mathcal{P}(\cdot|s,a)}(\mathcal{H}\boldsymbol{Q}^*)(s', \boldsymbol{\omega}) \\
\text{(def. of } \mathcal{H}) \quad &= \boldsymbol{\omega}^\mathsf{T}\boldsymbol{r}(s, a) + \gamma \cdot \boldsymbol{\omega}^\mathsf{T}\mathbb{E}_{s' \sim \mathcal{P}(\cdot|s,a)}\arg_Q \sup_{a' \in \mathcal{A}, \boldsymbol{\omega}' \in \Omega} \boldsymbol{\omega}^\mathsf{T}\boldsymbol{Q}^*(s', a', \boldsymbol{\omega}') \\
\text{(linearity of exp. \& cancel } \boldsymbol{\omega}^\mathsf{T} \text{ and } \arg_Q) \quad &= \boldsymbol{\omega}^\mathsf{T}\boldsymbol{r}(s, a) + \gamma \cdot \mathbb{E}_{s' \sim \mathcal{P}(\cdot|s,a)} \sup_{a' \in \mathcal{A}, \boldsymbol{\omega}' \in \Omega} \boldsymbol{\omega}^\mathsf{T}\boldsymbol{Q}^*(s', a', \boldsymbol{\omega}') \\
\text{(insert eq. (20), def. of } \boldsymbol{Q}^*) \quad &= \boldsymbol{\omega}^\mathsf{T}\boldsymbol{r}(s, a) + \gamma \cdot \mathbb{E}_{s' \sim \mathcal{P}(\cdot|s,a)} \sup_{a' \in \mathcal{A}, \boldsymbol{\omega}' \in \Omega} \boldsymbol{\omega}^\mathsf{T}\left\{ \arg_Q \sup_{\pi \in \Pi} \boldsymbol{\omega}'^\mathsf{T}\mathbb{E}_{\substack{\tau \sim (\mathcal{P}, \pi) \\ |s_0=s', a_0=a'}} \left[ \sum_{t=0}^{\infty} \gamma^t \boldsymbol{r}(s_t, a_t) \right] \right\} \\
\text{(use def. of } \arg_Q, \text{ explained below)} \quad &= \boldsymbol{\omega}^\mathsf{T}\boldsymbol{r}(s, a) + \gamma \cdot \mathbb{E}_{s' \sim \mathcal{P}(\cdot|s,a)} \sup_{a' \in \mathcal{A}} \boldsymbol{\omega}^\mathsf{T}\left\{ \arg_Q \sup_{\pi \in \Pi} \boldsymbol{\omega}^\mathsf{T}\mathbb{E}_{\substack{\tau \sim (\mathcal{P}, \pi) \\ |s_0=s', a_0=a'}} \left[ \sum_{t=0}^{\infty} \gamma^t \boldsymbol{r}(s_t, a_t) \right] \right\} \\
\text{(rearrange expectation and sup)} \quad &= \boldsymbol{\omega}^\mathsf{T}\boldsymbol{r}(s, a) + \gamma \cdot \boldsymbol{\omega}^\mathsf{T}\arg_Q \sup_{\pi \in \Pi} \boldsymbol{\omega}^\mathsf{T}\mathbb{E}_{\substack{\tau \sim (\mathcal{P}, \pi) \\ s_0 \sim \mathcal{P}(\cdot|s,a)}} \left[ \sum_{t=0}^{\infty} \gamma^t \boldsymbol{r}(s_t, a_t) \right] \\
\text{(merge 1st term to sum \& use def. of } \boldsymbol{Q}^* \text{ again)} \quad &= \boldsymbol{\omega}^\mathsf{T}\left\{ \arg_Q \sup_{\pi \in \Pi} \boldsymbol{\omega}^\mathsf{T}\mathbb{E}_{\substack{\tau \sim (\mathcal{P}, \pi) \\ |s_0=s, a_0=a}} \left[ \sum_{t=0}^{\infty} \gamma^t \boldsymbol{r}(s_t, a_t) \right] \right\} = \boldsymbol{\omega}^\mathsf{T}\boldsymbol{Q}^*(s, a, \boldsymbol{\omega})
\end{aligned}
$$

The fourth equation is due to a sandwich inequality, $\boldsymbol{\omega}^\mathsf{T} \arg_Q \sup_{\pi \in \Pi} \boldsymbol{\omega}^\mathsf{T} \boldsymbol{Q}^\pi \leq \sup_{\boldsymbol{\omega}' \in \Omega} \boldsymbol{\omega}^\mathsf{T} \arg_Q \sup_{\pi \in \Pi} \boldsymbol{\omega}'^\mathsf{T} \boldsymbol{Q}^\pi = \boldsymbol{\omega}^\mathsf{T} \arg_Q \sup_{\pi \in \Pi} \boldsymbol{\omega}_*^\mathsf{T} \boldsymbol{Q}^\pi = \boldsymbol{\omega}^\mathsf{T} \boldsymbol{Q}^{\pi'_{\boldsymbol{\omega}'_*}} \leq \boldsymbol{\omega}^\mathsf{T} \arg_Q \sup_{\pi \in \Pi} \boldsymbol{\omega}^\mathsf{T} \boldsymbol{Q}^\pi$, where $\boldsymbol{\omega}'_*$ and $\pi'_{\boldsymbol{\omega}'_*}$ are preference and policy corresponding to the supremums. According to the observation stated at the beginning, $d(\boldsymbol{Q}^*, \mathcal{T}\boldsymbol{Q}^*) = 0$. The preferred optimal value function is a fixed point of the proposed envelope version optimality operator. $\square$

Theorem 1 tells us the preferred optimal value function is one of the fixed-points of envelope optimality operator $\mathcal{T}$ in the value space. And we still need to show that this $\mathcal{T}$ is a contraction.

**Theorem 2** (Envelope Optimal Operator is a Contraction). *Let $\boldsymbol{Q}, \boldsymbol{Q}'$ be any two multi-objective Q-value functions in the envelope value space $\mathcal{Q}$ as defined above, the Lipschitz condition $d(\mathcal{T}\boldsymbol{Q}, \mathcal{T}\boldsymbol{Q}') \leq \gamma d(\boldsymbol{Q}, \boldsymbol{Q}')$ holds, where $\gamma \in [0, 1)$ is the discount factor of the underlying MOMDP $\mathcal{M}$ (see Section 2).*

*Proof.* Without the loss of generality, we assume $\sup_{a \in \mathcal{A}, \boldsymbol{\omega}' \in \Omega} \boldsymbol{\omega}^\mathsf{T} \boldsymbol{Q}(s, a, \boldsymbol{\omega}') \geq \sup_{a \in \mathcal{A}, \boldsymbol{\omega}' \in \Omega} \boldsymbol{\omega}^\mathsf{T} \boldsymbol{Q}'(s, a, \boldsymbol{\omega}')$ for some state $s$ and $\boldsymbol{\omega}$ of interest. Expand the expression of $d(\mathcal{T}\boldsymbol{Q}, \mathcal{T}\boldsymbol{Q}')$ we have

$$
\begin{aligned}
d(\mathcal{T}\boldsymbol{Q}, \mathcal{T}\boldsymbol{Q}')(s, a) &= \sup_{\substack{s \in \mathcal{S}, a \in \mathcal{A} \\ \boldsymbol{\omega} \in \Omega}} \left| \boldsymbol{\omega}^\mathsf{T}((\mathcal{T}\boldsymbol{Q})(s, a) - (\mathcal{T}\boldsymbol{Q}')(s, a)) \right| \\
&= \sup_{\substack{s \in \mathcal{S}, a \in \mathcal{A} \\ \boldsymbol{\omega} \in \Omega}} \left| \gamma \cdot \boldsymbol{\omega}^\mathsf{T} \mathbb{E}_{s' \sim P(\cdot|s,a)}(\mathcal{H}\boldsymbol{Q})(s', \boldsymbol{\omega}) - \gamma \cdot \boldsymbol{\omega}^\mathsf{T} \mathbb{E}_{s' \sim P(\cdot|s,a)}(\mathcal{H}\boldsymbol{Q}')(s', \boldsymbol{\omega}) \right| \\
&\leq \gamma \cdot \sup_{s' \in \mathcal{S}, \boldsymbol{\omega} \in \Omega} \left| \boldsymbol{\omega}^\mathsf{T} \left[ \arg_Q \sup_{a' \in \mathcal{A}, \boldsymbol{\omega}' \in \Omega} \boldsymbol{\omega}^\mathsf{T} \boldsymbol{Q}(s', a', \boldsymbol{\omega}') - \arg_Q \sup_{a'' \in \mathcal{A}, \boldsymbol{\omega}'' \in \Omega} \boldsymbol{\omega}^\mathsf{T} Q'(s', a'', \boldsymbol{\omega}'') \right] \right| \\
&\leq \gamma \cdot \sup_{s' \in \mathcal{S}, \boldsymbol{\omega} \in \Omega} \left| \sup_{a' \in \mathcal{A}, \boldsymbol{\omega}' \in \Omega} \boldsymbol{\omega}^\mathsf{T} \boldsymbol{Q}(s', a', \boldsymbol{\omega}') - \sup_{a'' \in \mathcal{A}, \boldsymbol{\omega}'' \in \Omega} \boldsymbol{\omega}^\mathsf{T} Q'(s', a'', \boldsymbol{\omega}'') \right|
\end{aligned}
$$

Step 2 to 3 is because $|\mathbb{E}[\cdot]| \leq \mathbb{E}[|\cdot|] \leq \sup |\cdot|$, and step 3 to 4 results from the cancellation between $\boldsymbol{\omega}^\mathsf{T}$ and $\arg_Q$ (as justified above). According to our assumption, let $a'$ and $\boldsymbol{\omega}'$ be the action and preference chosen to maximize the value of $\boldsymbol{\omega}^\mathsf{T}\boldsymbol{Q}$ for some state $s'$ and preference $\boldsymbol{\omega}$ of interest, then we derive

$$
\begin{aligned}
d(\mathcal{T}\boldsymbol{Q}, \mathcal{T}\boldsymbol{Q}')(s, a) &\leq \gamma \cdot \sup_{s' \in \mathcal{S}, \boldsymbol{\omega} \in \Omega} \left| \boldsymbol{\omega}^\mathsf{T} \boldsymbol{Q}(s', a', \boldsymbol{\omega}') - \sup_{a'' \in \mathcal{A}, \boldsymbol{\omega}'' \in \Omega} \boldsymbol{\omega}^\mathsf{T} \boldsymbol{Q}'(s', a'', \boldsymbol{\omega}'') \right| \\
&= \gamma \cdot \sup_{s' \in \mathcal{S}, \boldsymbol{\omega} \in \Omega} | \boldsymbol{\omega}^\mathsf{T} \boldsymbol{Q}(s', a', \boldsymbol{\omega}') - \boldsymbol{\omega}^\mathsf{T} \boldsymbol{Q}'(s', a', \boldsymbol{\omega}') \\
&\quad + \boldsymbol{\omega}^\mathsf{T} \boldsymbol{Q}'(s', a', \boldsymbol{\omega}') - \sup_{a'' \in \mathcal{A}, \boldsymbol{\omega}'' \in \Omega} \boldsymbol{\omega}^\mathsf{T} \boldsymbol{Q}'(s', a'', \boldsymbol{\omega}'') | \\
&\leq \gamma \cdot \sup_{s' \in \mathcal{S}, \boldsymbol{\omega} \in \Omega} \left| \boldsymbol{\omega}^\mathsf{T} \boldsymbol{Q}(s', a', \boldsymbol{\omega}') - \boldsymbol{\omega}^\mathsf{T} \boldsymbol{Q}'(s', a', \boldsymbol{\omega}') \right| \\
&\leq \gamma \cdot \sup_{\substack{s' \in \mathcal{S}, a' \in \mathcal{A} \\ \boldsymbol{\omega} \in \Omega}} \left| \boldsymbol{\omega}^\mathsf{T} \boldsymbol{Q}(s', a', \boldsymbol{\omega}') - \boldsymbol{\omega}^\mathsf{T} \boldsymbol{Q}'(s', a', \boldsymbol{\omega}') \right| = \gamma d(\boldsymbol{Q}, \boldsymbol{Q}')
\end{aligned}
$$

The step 2 to 3 arises from the w.l.o.g. assumption that $\boldsymbol{\omega}^\mathsf{T} \boldsymbol{Q}(s', a', \boldsymbol{\omega}') - \sup_{a'', \boldsymbol{\omega}''} \boldsymbol{\omega}^\mathsf{T} \boldsymbol{Q}'(s', a'', \boldsymbol{\omega}'') \geq 0$, as stated in lines 612 and 615. Thus, the whole expression in $|\cdot|$ is nonnegative and $\boldsymbol{\omega}^\mathsf{T} \boldsymbol{Q}(s', a', \boldsymbol{\omega}') - \boldsymbol{\omega}^\mathsf{T} \boldsymbol{Q}'(s', a', \boldsymbol{\omega}') \geq 0$ . We can discard the last two terms since $\boldsymbol{\omega}^\mathsf{T} \boldsymbol{Q}'(s', a', \boldsymbol{\omega}') \leq \sup_{a'' \in \mathcal{A}, \boldsymbol{\omega}'' \in \Omega} \boldsymbol{\omega}^\mathsf{T} \boldsymbol{Q}'(s', a'', \boldsymbol{\omega}'')$. Step 3 to 4 is because $\sup_{s', \boldsymbol{\omega}'} f(s', a', \boldsymbol{\omega}') \leq \sup_{s', a'', \boldsymbol{\omega}'} f(s', a'', \boldsymbol{\omega}')$ holds for any $a'$ and $f(\cdot)$.

This completes our proof that $\mathcal{T}$ is a contraction. $\square$

Remember that in our design, envelope version value distance $d$ is a pseudo-metric. In a pseudo-metric space iteratively applying contraction may not shrink to the desired fixed point. To assert the convergence effectiveness of our design for optimality operator, we need to investigate a generalized Banach's Fixed-Point Theorem in the pseudo-metric space.

### A.2.2 Generalized Banach's Fixed-Point Theorem

**Theorem 3** (Generalized Banach Fixed-Point Theorem). *Given that $\mathcal{T}$ is a contraction mapping with Lipschitz coefficient $\gamma$ on the complete pseudo-metric space $\langle \mathcal{Q}, d \rangle$, and $\boldsymbol{Q}^*$ is defined as that in Theorem 1, it is always true that $\lim_{n \to \infty} d(\mathcal{T}^n \boldsymbol{Q}, \boldsymbol{Q}^*) = 0$ for any $\boldsymbol{Q} \in \mathcal{Q}$.*

*Proof.* By the symmetry and triangle inequality of pseudo-metric, for any $\boldsymbol{Q}, \boldsymbol{Q}' \in \mathcal{Q}$,

$$
\begin{aligned}
d(\boldsymbol{Q}, \boldsymbol{Q}') &\leq d(\boldsymbol{Q}, \mathcal{T}\boldsymbol{Q}) + d(\mathcal{T}\boldsymbol{Q}, \mathcal{T}\boldsymbol{Q}') + d(\mathcal{T}\boldsymbol{Q}', \boldsymbol{Q}') \\
&\leq d(\boldsymbol{Q}, \mathcal{T}\boldsymbol{Q}) + \gamma d(\boldsymbol{Q}, \boldsymbol{Q}') + d(\mathcal{T}\boldsymbol{Q}', \boldsymbol{Q}') \\
\Rightarrow d(\boldsymbol{Q}, \boldsymbol{Q}') &\leq [d(\mathcal{T}\boldsymbol{Q}, \boldsymbol{Q}) + d(\mathcal{T}\boldsymbol{Q}', \boldsymbol{Q}')]/(1 - \gamma)
\end{aligned}
$$

Consider two points $\mathcal{T}^\ell \boldsymbol{Q}, \mathcal{T}^m \boldsymbol{Q}$ in the sequence $\{\mathcal{T}^n \boldsymbol{Q}\}$. Their distance is bounded by

$$
\begin{aligned}
d(\mathcal{T}^\ell \boldsymbol{Q}, \mathcal{T}^m \boldsymbol{Q}) &\leq [d(\mathcal{T}^{\ell+1}\boldsymbol{Q}, \mathcal{T}^\ell \boldsymbol{Q}) + d(\mathcal{T}^{m+1}\boldsymbol{Q}, \mathcal{T}^m \boldsymbol{Q})]/(1 - \gamma) \\
&\leq [\gamma^\ell d(\mathcal{T}\boldsymbol{Q}, \boldsymbol{Q}) + \gamma^m d(\mathcal{T}\boldsymbol{Q}, \boldsymbol{Q})]/(1 - \gamma) \\
&\leq \frac{\gamma^\ell + \gamma^m}{(1 - \gamma)} d(\mathcal{T}\boldsymbol{Q}, \boldsymbol{Q})
\end{aligned}
$$

since $\gamma \in [0, 1)$ the distance $d(\mathcal{T}^\ell \boldsymbol{Q}, \mathcal{T}^m \boldsymbol{Q})$ converge to 0 as $\ell, m \to \infty$, proving that $\{\mathcal{T}^n \boldsymbol{Q}\}$ is a Cauchy sequence. Because $\langle \mathcal{Q}, d \rangle$ is a complete pseudo-metric space, $\lim_{n \to \infty} d(\mathcal{T}^n \boldsymbol{Q}, \boldsymbol{Q}^\diamond) = 0$ for some $\boldsymbol{Q}^\diamond \in \mathcal{Q}$. Therefore,

$$
d(\mathcal{T}\boldsymbol{Q}^\diamond, \boldsymbol{Q}^\diamond) = \lim_{n \to \infty} d(\mathcal{T}^{n+1}\boldsymbol{Q}, \boldsymbol{Q}^\diamond) = \lim_{n \to \infty} d(\mathcal{T}^n \boldsymbol{Q}, \boldsymbol{Q}^\diamond) = 0
$$

We claim that $\boldsymbol{Q}^\diamond$ and $\boldsymbol{Q}^*$ must lie in the same equivalent class partitioned by relation $d(\cdot, \cdot) = 0$. Suppose $d(\boldsymbol{Q}^\diamond, \boldsymbol{Q}^*) \neq 0$, then we can get a contradiction

$$
\begin{aligned}
d(\boldsymbol{Q}^\diamond, \boldsymbol{Q}^*) &= d(\boldsymbol{Q}^\diamond, \mathcal{T}\boldsymbol{Q}^\diamond) + d(\mathcal{T}\boldsymbol{Q}^\diamond, \mathcal{T}\boldsymbol{Q}^*) + d(\mathcal{T}\boldsymbol{Q}^*, \boldsymbol{Q}^*) \\
&\leq 0 + \gamma d(\boldsymbol{Q}^\diamond, \boldsymbol{Q}^*) + 0 \\
&< d(\boldsymbol{Q}^\diamond, \boldsymbol{Q}^*)
\end{aligned}
$$

This proves our claim. Therefore, $\lim_{n \to \infty} d(\mathcal{T}^n \boldsymbol{Q}, \boldsymbol{Q}^*) = 0$ for any $\boldsymbol{Q} \in \mathcal{Q}$. $\square$

In other words, Theorem 3 guarantees that iteratively applying optimal operator $\mathcal{T}$ on any multi-objective Q-value function, the algorithm will terminate with a function $\boldsymbol{Q}^\diamond$ which is equivalent to $\boldsymbol{Q}^*$ under the measurement of pseudo-metric $d$. Actually, these $\boldsymbol{Q}^\diamond$'s are as good as $\boldsymbol{Q}^*$, since they all have the same utilities for each $\boldsymbol{\omega}$, and only differ in the real value when the utility corresponds a recess in the utility control frontier.

### A.2.3 Updating Scheme

**Hindsight Experience Replay (HER)** Paper [11] presents a technique for training a reinforcement learning to serve multiple goals. For each episode, the agent performs following a policy according to a randomly sampled goals. When updating, the agent uses the past trajectory update for multiple other goals in parallel. They referred this method as *hindsight experience replay* (HER). Though our settings are completely different, a similar method can be employed to update utility-based multi-objective Q-network here.

In the learning phase of the unknown linear preference scenario, for each training episode, the MORL agent randomly sample a preference $\boldsymbol{\omega}$ from a certain distribution $\mathcal{D}_\omega$. When updating the multi-objective Q-network accordingly, for each sampled transition record $(s_t^i, a_t^i, \boldsymbol{r}_t^i, s_{t+1}^i)$ from replay buffer $\mathcal{D}_\tau$, we associate it with $N_\omega$ preferences $\{\boldsymbol{\omega}_1, \boldsymbol{\omega}_2, \ldots, \boldsymbol{\omega}_{N_\omega}\}$ sampled from $\mathcal{D}_\omega$. The update is applied to an expended batch of size `minibatch_size` $\times$ `N`$_\omega$. Note that the preferences sampled for actions only influence the agent's actions, not the environment dynamics. In this way, the trajectories can be replayed with arbitrary preferences with "hindsight".

**Homotopy Optimization** We use deep neural networks to approximate bounded functions in $\mathcal{Q} \subseteq (\Omega \to \mathbb{R}^m)^{\mathcal{S} \times \mathcal{A}}$ with parameters $\theta$. We refer to this neural network as a *multi-objective Q-network*. To drag $\boldsymbol{Q}$ close to $\mathcal{T}\boldsymbol{Q}$ at each update step, the multi-objective Q-network can be trained by minimizing a series of loss functions

$$
L_k^{\mathbb{A}}(\theta) = \mathbb{E}_{s,a,\boldsymbol{\omega}}\left[\|\boldsymbol{y}_k - \boldsymbol{Q}(s, a, \boldsymbol{\omega}; \theta)\|_2^2\right], \tag{21}
$$

which changes at each iteration $k$, where $\boldsymbol{y}_k = \mathbb{E}_{s'}\left[\boldsymbol{r}(s, a) + \gamma(\mathcal{H}\boldsymbol{Q})(s', a', \boldsymbol{\omega}; \theta_k)\right]$ is the target of iteration $k$. The target is fixed during optimizing this loss function.

Minimizing the loss function $L^{\mathbb{A}}$ is trying to drag the vector $\boldsymbol{Q}$ close to $\mathcal{T}\boldsymbol{Q}$. This ensures the correctness of our algorithm, to predict a $\boldsymbol{Q}$ as the real solution, while this means the square error is hard to be optimized in practice. To address this problem, we use a sequence of auxiliary loss function $L^{\mathbb{B}}$ to directly optimized the value metric $d$, which is defined by

$$
L_k^{\mathbb{B}}(\theta) = \mathbb{E}_{s,a,\boldsymbol{\omega}}[|\boldsymbol{\omega}^{\mathsf{T}}\boldsymbol{y}_k - \boldsymbol{\omega}^{\mathsf{T}}\boldsymbol{Q}(s, a, \boldsymbol{\omega}; \theta)|] \tag{22}
$$

Our final loss function sequence $L_k(\theta) = (1 - \lambda_k) \cdot L_k^{\mathtt{A}}(\theta) + \lambda_k \cdot L_k^{\mathtt{B}}(\theta)$, where $\lambda_k$ is a weight to trade off between losses $L_k^{\mathtt{A}}$ and $L_k^{\mathtt{B}}$. We increase the value of $\lambda_k$ from 0 to 1, to shift our loss function from $L^{\mathtt{A}}$ to $L^{\mathtt{B}}$. This method called *homotopy optimization* [12] is effective since for each update step, it uses the optimization result from the previous step as the initial guess. In the envelope deep MORL algorithm, $L^{\mathtt{A}}$ first ensure the prediction of $\boldsymbol{Q}$ is close to any real expected total reward, though it is hard to be optimal. And then $L^{\mathtt{B}}$ can provide an auxiliary force to pull the current guess along the direction with better utility. Figure 6 illustrate an explanation for this homotopy optimization.

The parameters of the multi-objective Q-network will be updated by $\theta_{k+1} \leftarrow \theta_k - \eta$, where

$$\eta \propto \nabla_{\theta=\theta_k} L_k(\theta) = -\mathbb{E}_{s,a,s'} \left[ \left( \boldsymbol{r} + \gamma(\mathcal{H}\boldsymbol{Q})(s', a', \boldsymbol{\omega}; \theta_k) - \boldsymbol{Q}(s, a, \boldsymbol{\omega}; \theta_k) \right)^{\mathsf{T}} \nabla_{\theta=\theta_k} \boldsymbol{Q}(s, a, \boldsymbol{\omega}; \theta) \right]. \quad (23)$$

**Figure 6:** An explanation for homotopy optimization method used in the envelope deep MORL algorithm. The MSE loss $L^{\mathtt{A}}$ is hard for optimization since there are many local minima over its landscape. Although the value metric loss $L^{\mathtt{B}}$ has fewer local minima, it is also hard for optimization since there are many vectors $\boldsymbol{Q}$ minimizing value metric $d$. The landscape of $L^{\mathtt{B}}$ is too flat. The homotopy path connecting $L^{\mathtt{A}}$ and $L^{\mathtt{B}}$ provides better opportunities to find the global optimal parameters $\theta^*$

When updating, we sample a minibatch of transition records from this replay buffer with HER. Theorems 1-3 and 5 guarantees the convergence of this minibatch updating, with an extra assumption that we can update the Q-function according to equation 23 for each $\boldsymbol{\omega} \in \Omega$ infinite times. We use *hindsight experience replay* (HER) to ensure this. Notice that we will apply our optimality filter on the HER expended batch. Therefore the cost of solving the convex envelope is acceptable. Our multi-objective Q-network can also be replaced with other models similar to those in single-objective off-policy RL algorithms. In the experiment, we also use some popular deep reinforcement learning techniques to stabilize and speed up our algorithms. The skeleton of our envelope deep MORL is shown as Algorithm 1.

# B    Experimental Details

We first demonstrate our experimental results on two synthetic domains, Deep Sea Treasure (DST) and Fruit Tree Navigation (FTN), as well as two complex real domains, Task-Oriented Dialog Policy Learning (Dialog) and SuperMario Game (SuperMario). We also elaborate specific model architecture and and implementation details in this section.

## B.1    Domain Details

**Deep Sea Treasure (DST)**    Our first experiment domain is a grid-world navigation problem, *Deep Sea Treasure*. This episodic problem was first explicitly created to highlight the limitations of linear scalarization [14] . However, in this paper, we use this environment as a delayed linear preference scenario. We ensure the Pareto frontier of this environment is convex, therefore the Pareto frontier itself is the its CCS.

In DST, an agent controls a submarine searching for treasures in a $10 \times 11$-grid world while trading off `time-cost` and `treasure-value`. The grid world contains 10 treasures of different values. Their values increase as their distances from the starting point $s_0 = (0, 0)$ increase. An agent's action spaces are formed by navigation in four directions. The reward has two dimensions: the first dimension indicates a time penalty, which is $-1$ on all turns; and the second dimension is the treasure value which is 0 except when the agent moves into a treasure location. We ensure the Pareto frontier of this environment to be convex. We depicted the map in Figure 7.

**Figure 8:** Fruit Tree Navigation (FTN): An agent travels from the root node to one of the leaf node to pick a fruit according to a post-assigned preference $\boldsymbol{\omega}$ on the components of nutrition, treated as different objectives. The observation of an agent is its current coordinates (`row`, `col`), and its valid actions are moving to the left or the right subtree.

**Fruit Tree Navigation (FTN)** Our second experiment domain is a full binary tree of depth $d$ with randomly assigned vectorial reward $\boldsymbol{r} \in \mathbb{R}^6$ on the leaf nodes. These rewards encode the amounts of six different components of nutrition of the *fruits* on the tree: {`Protein`, `Carbs`, `Fats`, `Vitamins`, `Minerals`, `Water`}. For every leaf node, $\exists \boldsymbol{\omega}$ for which its reward is optimal, thus all leaves lie on the CCS. The goal of our MORL agent is to find a path from the root to a leaf node that maximizes utility for a given preference, choosing between left or right subtrees at every non-terminal node.

Figure 8 shows an instance of the *fruit tree navigation* task when $d = 6$, in which every non-leaf node is associated with zero reward and every fruit is a potential optimal solution in the convex cover set of the Pareto frontier. To construct this, we sample $\boldsymbol{r}^{(i)} = (\boldsymbol{v}_+^{(i)} + \boldsymbol{v}_-^{(i)})/\|\boldsymbol{v}^{(i)}\|_2$, where $\boldsymbol{v}^{(i)} \sim \mathcal{N}_6(\boldsymbol{0}, \boldsymbol{I})$, for each fruit $i$ on a leaf node. The optimal multiple-policy model for this tree structured MOMDP should contain all the paths from the root to different desired fruits. In experiments, we also test on $d = 5$ and $d = 7$ cases.

**Figure 7:** Deep Sea Treasure (DST): An agent controls a submarine searching for treasures in a $10 \times 11$-grid world. The state $s_t$ consists of the agent's current coordinates $(x, y)$. An agent's action spaces is navigation in four directions. The reward received by the agent is a 2-dimensional vector (time penalty, treasure value).

In this multi-objective environment, an optimal policy can be easily learned if we know the preference function $f_{\boldsymbol{\omega}}(\cdot) = \langle \boldsymbol{\omega}, \cdot \rangle$ for scalarization. However, since here we are interested in evaluating whether a multiple-policy neural network, trained with deep MORL algorithms, can find and maintain all the potential optimal policies (i.e., paths to every leaf node) when the preference function is unknown, and adapt to the optimal policy when a specific preference is given or hidden during execution.

**Task-Oriented Dialog Policy Learning (Dialog)** Our third experimental domain is a modified task-oriented dialog system in the restaurant reservation domain based on PyDial [36], where an agenda-base user simulator [43] with an error model to simulate the recognition and understanding errors arisen in the real system due to in the speech noise and ambiguity. A. We consider the task success rate and the dialog brevity (measured by number of turns) as two competing objectives of this domain.

Finding a good trade-off between multiple potentially competing objectives is usually domain-specific and not straightforward. For example, in the case when the objectives are brevity and success, if the relative importance weight for success is too high, the resulting policy is insensitive to potentially annoying actions such as `repeat()` provided that the dialogue is eventually successful. In this case, the obtained optimal policy cannot fit all users' preference, and sometimes is out of our expectation. Adaptation to user preferences and balancing these objectives is rarely considered.

In the standard single-objective reinforcement learning formulation, the goal of the policy model is to interact with a human user by choosing actions in each turn to maximize future rewards. We define the dialogue state shared by dialogue state tracker in the $t$-th turn as state $s_t$. The action taken by policy model under current

policy $\pi_\theta$ with parameters $\theta$ in the $t$-th turn as $a_t$, and $a_t \sim \pi(\cdot|s_t)$. The stochastic transition kernel is unknown but determined by human users or user simulators. In an ideal dialogue environment, once the policy model emits an action $a_t$, the human user will give an explicit feedback, like a normal response or a feedback of whether the dialogue is successful, which will be converted to a reward signal $r_t$ delivering to the policy model immediately, and then the policy model will transit to next state $s_t$. The reward signal is an average of values of two objectives, brevity and success, e.g., $r_t = 0.5r_t^{\text{turn}} + 0.5r_t^{\text{succ}}$. Typically, $r_t^{\text{turn}}$ is fixed for each turn as a negative constant $R^{\text{turn}}$, while $r_t^{succ}$ equals a positive constant $R^{\text{succ}}$ only when the dialogue terminates and receives a successful user feedback otherwise it equals zero.

To address this problem, we transform the dialogue learning process into a MORL scenario with vectorized (2-D) rewards: $\boldsymbol{r}_t = \begin{bmatrix} r_t^{\text{turn}} & r_t^{\text{succ}} \end{bmatrix}^\mathsf{T}$, where $r_t^{\text{turn}}$ is a turn penalty for the brevity objective and $r_t^{\text{succ}}$ is a reward provided on successful completion of the task. In the learning phase, the linear preference $\boldsymbol{\omega}$ over these two objectives are unknown, while the computational resources are abundant. The task-oriented dialogue system needs to learn all the possible optimal policies with sampled $\boldsymbol{\omega}$'s (achieved by user simulators or collected interactions with real users). While in the adaptation phase, learning is unaffordable because of the limitation of resources. The task-oriented dialogue system needs to respond the user with a specified user preference. User's utility increase is aligned with the system's utility increase $\boldsymbol{\omega}^\mathsf{T}\boldsymbol{r}_t$. Paper [44] proposes a structured method, which is equivalent to the scalarized baseline without hindsight experience replay, for finding the optimal weights for a multi-objective reward function.

As for more experimental details, our dialog domain is a restaurant reservation hotline which provides information about restaurants in Cambridge. There are 3 search constraints, 9 informational items that the user can request, and 110 database entities. The reward $R^{\text{turn}}$ is $-1$ for each turn, and $R^{\text{succ}} = 20$. The maximal length of dialogue is 25. We apply our envelope deep MORL algorithm to this dialogue policy learning task, and compare to traditional single-objective methods and other baselines. All the single-objective and multi-objective reinforcement learning are trained for 3,000 sessions with 15% simulated speech recognition and understanding error rate.

**Multi-Objective SuperMario Game (SuperMario)** Our final environment is a version of the popular video game Super Mario Bros. We modify the open-source environment from OpenAI gym [37] to provide vectorized rewards encoding five different objectives: `x-pos`: value corresponding to the difference in Mario's horizontal position between current and last time point, `time`: a small negative time penalty, `deaths`: a large negative penalty given each time Mario dies[3], `coin`: rewards for collecting coins, and `enemy`: rewards for eliminating an enemy. The state is a stack of four continuous frames of game images rendered by the simulator, and there are seven valid actions each step: {'NOOP','right','right+A','right+B','right+A+B', 'A', 'left'}, where the button 'A' is used to jump and the button 'B' is used to run. We restrict the Mario to only play the stage I.

We use an A3C [38] variant of our envelope MORL algorithm. During the learning phase, the agent does not know the underlying preference, and hence needs to learn a multi-objective policy within 32k training episodes. During the adaptation phase, we test our agents under 500 uniformly random preferences and test the its preference elicitation ability (as described in Section 3) within 100 episodes to uncover the underlying preference that maximizes utility.

### B.2 Implementation Details

Our multi-objective Q-network can be replaced with any model similar to that in single-objective off-policy RL algorithms like DDPG [45], NAF [46] or SDQN [47]. In the experiment, we use a variate of deep reinforcement learning techniques including double Q-learning [48] with a target network and prioritized experience replay [49], which stabilize and speed up our algorithms.

**Architectures of the Multi-objective Q-Network** We implement the Multi-objective Q-networks (MQNs) by 4 fully connected hidden layers with $\{16, 32, 64, 32\} \times (\texttt{dim}(S) + m)$ hidden unites respectively. The multi-objective Q-networks are similar to Deep Q-Networks (DQNs) [34], but differs on inputs. An input of the multi-objective Q-network is a concatenation of state representation and parameters of a linear preference function. The output layer of the scalarized MORL algorithm is of size $|\mathcal{A}|$, and that of envelope version is of size $m \times |\mathcal{A}|$. Here $\texttt{dim}(\mathcal{S})$ is the dimensionality of the state space, $|\mathcal{A}|$ is the cardinality of the action set, and $m$ is the number of objectives.

**Multi-Objective A3C** We use the multi-objective A3C (MoA3C) algorithms for Mario experiment. The skeleton of the envelope MoA3C algorithm is provided in Algorithm 2. Im MoA3C Both critic and actor networks contain three shared convolutional layers for feature extraction from raw images input. The extracted features are then concatenated with preferences, and fed into two-layer fully connected networks for output. For the scalarized version MoA3C, the output of the critic network is just one-dimensional utility prediction, whereas the output of the envelope version critic network is $m$-dimensional returns prediction. Both scalarized

**Figure 9:** Quantitative evaluation metrics for multi-objective reinforcement learning. (a.) Coverage ratio measures an agent's ability to find all the potential optimal solutions in the convex coverage set of Pareto frontier. (b.) Adaptation quality measures an agent's ability of policy adaptation to real-time specified preferences.

and envelope versions have the same actor network architecture to output the probability distribution over the action space. We train them with 16 workers in parallel with different sampled preferences, and it take around 10 hours for the envelope version MoA3C to converge to a good level of performance.

**Training with Prioritized Double Q-Learning**    When training the with our MORL algorithm on DST and FTN tasks, we employ techniques of *prioritized experience reply* [49] and *double Q-learning* [48] to speed up the training process and to yield more accurate value estimates. Double Q-Learning introduces a target network $Q_{\text{target}}$ to replace the estimate of $\mathcal{T}Q(s, a, \boldsymbol{\omega})$, $y_t(\boldsymbol{\omega}) = \boldsymbol{\omega}^\intercal \boldsymbol{r} + \gamma Q_{\text{target}}(s', a, \boldsymbol{\omega})$ with $y_t^{\text{double}}(\boldsymbol{\omega}) = \boldsymbol{\omega}^\intercal \boldsymbol{r} + \gamma Q_{\text{target}}(s', \arg\max_a Q(s', a, \boldsymbol{\omega}), \boldsymbol{\omega})$ for scalarized version of algorithm, and similarly we replace $y_t(\boldsymbol{\omega}) = \boldsymbol{\omega}^\intercal \boldsymbol{r} + \gamma \boldsymbol{\omega}^\intercal \boldsymbol{Q}_{\text{target}}(s', a, \boldsymbol{\omega})$ with $y_t^{\text{double}}(\boldsymbol{\omega}) = \boldsymbol{\omega}^\intercal \boldsymbol{r} + \gamma \boldsymbol{\omega}^\intercal \boldsymbol{Q}_{\text{target}}(s', \arg\max_a \boldsymbol{\omega}^\intercal \boldsymbol{Q}(s', a, \boldsymbol{\omega}), \boldsymbol{\omega})$ for envelope version of MORL algorithm. We update the target network by coping from Q-network every 100 steps. The priority of sampling transition $\tau_i = (s, a, \boldsymbol{r}, s')$ is $p_i^{\text{scalarized}} = |y^{\text{double}}(\boldsymbol{\omega}) - Q(s, a, \boldsymbol{\omega})|$ for the scalarized version of MORL algorithm, and similarly $p_i^{\text{envelope}} = |y^{\text{double}}(\boldsymbol{\omega}) - \boldsymbol{\omega}^\intercal \boldsymbol{Q}(s, a, \boldsymbol{\omega})|$ for the envelope version of algorithm, where $\boldsymbol{\omega}$ is sampled from the distribution $\mathcal{D}_\omega$. When updating the network, a trajectory is sampled by $\tau_i \sim P(i) = p_i / \sum_i p_i$. The replay memory size is 4000 and the batch size is 32. For the deep tree navigation task, we train each model for total 5000 episodes, and update it by Adam optimizer every step after at least a batch experiences are stored in reply buffer, with a learning rate $\texttt{lr} = 0.001$.

**Training Details for Dialogue Policy Learning**    All the single-objective and multi-objective reinforcement learning are trained for 3,000 sessions. We evaluate learned policies on 5,000 sessions with randomly assigned user preferences. The preference distribution $\mathcal{D}_\omega$ is same as the one we used in previous deep sea treasure experiment (see Section 4.3.1), which is a nearly uniform distribution. For the single-objective reinforcement learning algorithms, we set three groups of $\boldsymbol{\omega}$ as $\{(0.5, 0.5), (0.2, 0.8), (0.8, 0.2)\}$. For the envelope deep MORL algorithms, the homotopy path is a monotonically increasing track where $\lambda$ increases from $0.0$ to $1.0$ exponentially. The number of sampled preferences $N_\omega$ is 32 for both scalarized and envelope deep MORL algorithms. The exploration policy used for training these reinforcement learning algorithms is $\epsilon$-greedy, where $\epsilon = 0.5$ initially and then decays to zero linearly during the training process. For all the single-objective and multi-objective algorithms, we employ the same deep Q-network architecture, which comprises 3 fully connected hidden layers with $\{16, 32, 32\} \times (\texttt{dim}(S) + m)$ hidden units. The minibatch size is 64 for all. An Adam optimizer is used for updating the parameters of all these algorithms with an initial learning rate $\texttt{rl} = 0.001$.

**Computing Infrastructure**    We ran the synthetic experiments and the dialog experiments on a workstation with one GeForce GTX TITAN X GPU, 12 Intel(R) Core(TM) i7-5820K CPUs @ 3.30GHz, and 32G memory and ran the SuperMario experiments on a cluster with twenty 2080 RTX GPUs, 40 CPUs and 200GB memory.

# C    Additional Experimental Results

## C.1    Evaluation Metrics

We design two metrics to evaluate the empirical performance of our algorithms on test tasks. Slightly different from the main article, we introduce adaptation quality (AQ) here other than adaptation error to adjust the value range and get a score in $(0, 1]$.

**Coverage Ratio (CR).** The first metric is *coverage ratio* (CR), which evaluates the agent's ability to recover optimal solutions in the convex coverage set (CCS). If $\mathcal{F} \subseteq \mathbb{R}^m$ is the set of solutions found by the agent (via sampled trajectories), we define $\mathcal{F} \cap_\epsilon \texttt{CCS} := \{x \in \mathcal{F} \mid \exists y \in \texttt{CCS} \text{ s.t. } \|x - y\|_1 / \|y\|_1 \leq \epsilon\}$ as the intersection between these sets with a tolerance of $\epsilon$. The CR is then defined as:

$$\texttt{CR}_{\texttt{F1}}(\mathcal{F}) = 2 \cdot \frac{\texttt{precision} \cdot \texttt{recall}}{\texttt{precision} + \texttt{recall}}, \tag{24}$$

**Algorithm 2:** Envelope Multi-Objective A3C (EMoA3C) Algorithm - Pseudocode for each actor-learner thread

---

**Input:**
- a preference sampling distribution $\mathcal{D}_\omega$;
- minibatch sizes for transitions $N_\tau$ and for preference $N_\omega$;
- a multi-objective critic-network $\boldsymbol{V}$ parameterized by $\theta_v$;
- an actor-network $\pi$ parameterized by $\theta_\pi$;
- a balance weight $\lambda$ for critic losses $L^{\texttt{A}}$ and $L^{\texttt{B}}$.

Initialize replay buffer $\mathcal{D}_\tau$.
**for** *episode* = $1, \ldots, M$ **do**

    Synchronize thread-specific parameters $\theta'_v = \theta_v$ and $\theta'_\pi = \theta_\pi$.
    Sample a linear preference $\boldsymbol{\omega} \sim \mathcal{D}_\omega$.
    **for** $t = 0, \ldots, N-1$ **do**

        Observe state $s_t$.
        Sample an action $a_t$ using according to policy $\pi(a_t|s_t, \boldsymbol{\omega}; \theta'_\pi)$.
        Receive a multi-objective reward $\boldsymbol{r}_t$ and observe new state $s_{t+1}$.
        Store transition $(s_t, a_t, \boldsymbol{r}_t, s_{t+1})$ in $\mathcal{D}_\tau$.
        **if** *update* **then**

            Sample random minibatch of transitions $(s_j, a_j, \boldsymbol{r}_j, s_{j+1})$ from $\mathcal{D}_\tau$.
            Sample $N_\omega$ preferences $W = \{\boldsymbol{\omega}_1, \boldsymbol{\omega}_2, \ldots, \boldsymbol{\omega}_{N_\omega}\} \sim \mathcal{D}_\omega$.
            Compute

$$(\mathcal{T}\boldsymbol{V})_{ij} = \begin{cases} \boldsymbol{r}_j \cdot & \text{for terminal } s_{j+1}; \\ \boldsymbol{r}_j + \gamma \arg_V \max_{a \in \mathcal{A}, \boldsymbol{\omega}' \in W} \boldsymbol{\omega}_i^{\mathsf{T}} \boldsymbol{V}(s_{j+1}, \boldsymbol{\omega}'; \theta), & \text{for non-terminal } s_{j+1}. \end{cases}$$

            Calculate $d\theta_v$ according to similar equations 6 and 7 w.r.t. $\theta'_v$:

$$d\theta_v = (1-\lambda) \cdot \nabla_{\theta'_v} L^{\texttt{A}}(\theta'_v) + \lambda \cdot \nabla_{\theta'_v} L^{\texttt{B}}(\theta'_v).$$

            Calculate $d\theta_\pi$ using the advantage w.r.t. $\theta'_v$ and $\theta'_\pi$

$$d\theta_\pi = \frac{1}{N_\omega N_\tau} \sum_{i,j} \nabla_{\theta'_\pi} \log \pi(a_j|s_j, \boldsymbol{\omega}_i; \theta'_\pi) \left[ \boldsymbol{\omega}_i^{\mathsf{T}} ((\mathcal{T}\boldsymbol{V})_{ij} - \boldsymbol{V}(s_j, \boldsymbol{\omega}_i; \theta'_v)) \right].$$

        Perform asynchronous update of $\theta_v$ using $d\theta_v$ and of $\theta_\pi$ using $d\theta_\pi$.

---

where the $\texttt{precision} = |\mathcal{F} \cap_\epsilon \texttt{CCS}|/|\mathcal{F}|$, indicating the fraction of optimal solutions among the retrieved solutions, and the $\texttt{recall} = |\mathcal{F} \cap_\epsilon \texttt{CCS}|/|\texttt{CCS}|$, indicating the fraction of optimal instances that have been retrieved over the total amount of optimal solutions (see Figure 3(a)). The F1 score is their harmonic mean. In our evaluation of both synthetic tasks DST and FTN, we set $\epsilon = 0.00$ for $\mathcal{F}_{\Pi_{\mathcal{L}}}$ (executive frontier) and $\epsilon = 0.20$ for $\mathcal{F}_Q$ (frontier predicted by Q-function). Figure 9 (a.) illustrates an example of the computation of coverage ratio.

**Adaptation Quality (AQ).** Our second metric compares the retrieved control frontier with the optimal one, when an agent is provided with a specific preference $\boldsymbol{\omega}$ during the adaptation phase. The adaptation quality is defined by

$$\texttt{AQ}(\mathcal{C}) = \frac{1}{1 + \alpha \cdot \texttt{err}_{\mathcal{D}_\omega}}, \tag{25}$$

where the $\texttt{err}_{\mathcal{D}_\omega} = \mathbb{E}_{\boldsymbol{\omega} \sim \mathcal{D}_\omega}[|\mathcal{C}(\boldsymbol{\omega}) - \mathcal{C}_{\text{opt}}(\boldsymbol{\omega})|/\mathcal{C}_{\text{opt}}(\boldsymbol{\omega})]$ is the expected relative error between optimal control frontier $\mathcal{C}_{\text{opt}} : \Omega \to \mathbb{R}$ with $\boldsymbol{\omega} \mapsto \max_{\hat{\boldsymbol{r}} \in \texttt{CCS}} \boldsymbol{\omega}^{\mathsf{T}} \hat{\boldsymbol{r}}$ and the agent's control frontier $\mathcal{C}_{\pi_{\boldsymbol{\omega}}} = \boldsymbol{\omega}^{\mathsf{T}} \hat{\boldsymbol{r}}_{\pi_{\boldsymbol{\omega}}}$, and $\alpha$ is a scaling coefficient to amplify the discrepancy.

Similarly, $\mathcal{C}_Q$ is the control frontier guessed by an agent (via predictions from Q-network) and we can compute the predictive $\texttt{AQ}(\mathcal{C}_Q)$ to evaluate the quality of multi-objective Q-network on the value prediction accuracy.

In all experiment domains, we use Gaussian distributions which are restricted to be positive part and $\ell 1$-normalized as our $\mathcal{D}_\omega$. We set $\alpha = 0.01$ for the DST task (because the penalty range is large), and $\alpha = 10.0$ for the FTN task (because the value differences are small). Figure 9 (b.) shows examples of optimal control frontier, retrieved control frontier, and the control discrepancy. Overall, CR provides an indication of agent's ability to learn the space of optimal policies in the learning phase, while AE tests its ability to adapt to new scenarios.

## C.2 Deep Sea Treasure (DST)

We show more experimental results on DST tasks in this section. We train all agent for 2000 episodes. After training in the learning phase, our envelope MORL algorithms find all the potential optimal solutions and their corresponding policies.

Figure 10 presents the real CCS and the retrieved solutions of a MORL algorithm. The scalarized and envelope algorithm can find all the whole CCS. Figure 10 (b.) illustrates the real control frontier (the blue curve), retrieved control frontier (the green curve), and the predicted control frontier (the orange line). The retrieved control frontier is almost overlapped with the real control frontier, which indicates that the alignment between preferences and optimal policies is perfectly well. The agent can respond any given preference with the policy resulting in best utility.

**Figure 10:** The solutions and control frontier found by MORL algorithm in the deep sea treasure task. (a.) The real CCS and the retrieved solutions. (b.) The real control frontier, predicted control frontier, and retrieved control frontier in the adaptation phase.

| Method | DST | | |
|---|---|---|---|
| | CR F1 | Exe-AQ ($\alpha = 0.01$) | Pred-AQ ($\alpha = 0.01$) |
| MOFQI | $0.639 \pm 0.421$ | $0.417 \pm 0.134$ | $0.226 \pm 0.138$ |
| CN+OLS | $0.751 \pm 0.163$ | $0.743 \pm 0.008$ | $0.177 \pm 0.089$ |
| Scalarized | $0.989 \pm 0.024$ | $0.998 \pm 0.001$ | $\mathbf{0.950 \pm 0.034}$ |
| Envelope | $\mathbf{0.994 \pm 0.001}$ | $\mathbf{0.998 \pm 0.000}$ | $0.850 \pm 0.045$ |

**Table 4:** Performance comparison of different MORL algorithm in learning and adaptation phases on the DST environment. The `Exe-AQ` is the AQ measured on the real trajectories in the adaptation phase, and the `Pred-AQ` is the AQ values measured on the predictions of Q functions.

Table 4 provides the coverage ratio (CR) and adaptation quality (AQ) comparisons of different MORL algorithms. We trained all the algorithm in 2000 episodes and test for another 2000 episodes. Each data point in the table is an average of 5 train and test trails. For the CN+OLS baseline, we allow it to iterate for 25 corner weights. The envelope algorithm achieves best CR and execution AQ, and the scalarsized algorithm achieves best predictive adaption quality. Note that traditional evaluations rarely test the algorithm's ability in learning phase and adaptation phase separately. Thus our setting is more challenging.

The classical deep sea treasure task shows the effectiveness of our deep multi-objective reinforcement learning algorithms, while it is relatively easy for the agent to find all the good policies. It only contains 10 potentially optimal solutions in the real CCS, therefore scalarized algorithm can efficiently solve this problem.

## C.3 Fruit Tree Navigation (FTN)

**Figure 11:** Coverage Ratio (CR) and Adaptation Quality (AQ) comparison of the scalarized deep MORL algorithm and the envelope deep MORL algorithm tested on fruit tree navigation tasks of depths d = 5, 6, 7. Trained on 5000 episodes and test on 2000 and 5000 episode to estimate CR and AQ, respectively. Each data point in the figure is an average of 5 trails of training and test.

**Sample Efficiency**    To compare sample efficiency during the learning phase, we train envelope MORL algorithm and baselines on the FTN task of depth $5, 6, 7$ for 5000 episodes. We compute coverage ratio (CR) over 2000 episodes and adaptation quality (AQ) over 5000 episodes. Figure 11 (blue and orange curves for $d = 6$) shows plots for the metrics computed over a varying number of sampled preferences $N_\omega$ from 1 to 128. When $N_\omega = 1$, both algorithms only update under single preference each time, therefore runs fast while it makes wasteful use of interactions. When $N_\omega = 128$, both algorithm needs to update for 128 sampled preferences in a batch. In this case the algorithms run slowly, while can make better use of interactions. Each point on the curve is averaged over 5 experiments. We observe that the envelope MORL algorithm consistently has a better CR and AQ scores than the scalarized baseline, with smaller variances. As $N_\omega$ increases, CR and AQ both increase,

which shows better use of historical interactions for both algorithms when $N_\omega$ is larger. This reinforces our theoretical analysis that the envelope MORL algorithm has better sample efficiency than the scalarized baseline.

Table 5 compares their coverage ratios. Each entry in the table is an average of 5 experiments (train and test). Due to the property of FTN task that every solution is potentially optimal, the CR precision is always 1 for both scalarized and envelope algorithm. While for the recall, which indicates the ability to find unseen optimal solutions in the learning phase, the envelope deep MORL algorithm is better than the scalarized version for all numbers of sampled preferences, and has relatively lower variances. Therefore the same to F1 scores. As $N_\omega$ increases, the CR value increases for both scalarized version and envelope version algorithms, which verifies aa better use of historical interactions for both algorithm when $N_\omega$ is larger. As here the initial performance for the envelope version algorithm is good enough, it suddenly surpasses 0.98 of CR F1 score when it can sample more than one preference each update.

| $N_\omega$ | Execution AQ | | Prediction AQ | | CR F1 | |
|---|---|---|---|---|---|---|
| | Scalarized | Envelope | Scalarized | Envelope | Scalarized | Envelope |
| 1 | $0.7037 \pm 0.012$ | $0.759 \pm 0.066$ | $0.6474 \pm 0.054$ | $0.6915 \pm 0.105$ | $0.625 \pm 0.057$ | $0.924 \pm 0.051$ |
| 4 | $0.7701 \pm 0.026$ | $0.9101 \pm 0.006$ | $0.7216 \pm 0.034$ | $0.7853 \pm 0.049$ | $0.7654 \pm 0.077$ | $0.9856 \pm 0.004$ |
| 8 | $0.8205 \pm 0.023$ | $0.9261 \pm 0.015$ | $0.7714 \pm 0.02$ | $0.7675 \pm 0.033$ | $0.856 \pm 0.067$ | $0.9808 \pm 0.007$ |
| 16 | $0.8255 \pm 0.044$ | $0.9306 \pm 0.007$ | $0.8097 \pm 0.016$ | $0.8417 \pm 0.007$ | $0.8976 \pm 0.062$ | $0.9952 \pm 0.004$ |
| 32 | $0.8597 \pm 0.035$ | $0.9402 \pm 0.011$ | $0.7989 \pm 0.032$ | $0.8513 \pm 0.01$ | $0.914 \pm 0.044$ | $0.987 \pm 0.021$ |
| 64 | $0.877 \pm 0.031$ | $0.9506 \pm 0.001$ | $0.7778 \pm 0.032$ | $0.8497 \pm 0.018$ | $0.9452 \pm 0.02$ | $0.9904 \pm 0.007$ |
| 128 | $0.8705 \pm 0.03$ | $0.9536 \pm 0.002$ | $0.8081 \pm 0.04$ | $0.868 \pm 0.025$ | $0.9258 \pm 0.024$ | $0.9952 \pm 0.011$ |

**Table 5:** Sample Efficiency - Coverage Ratio (CR) and Adaptation Quality (AQ) comparison of the scalarized deep MORL algorithm and the envelope deep MORL algorithm tested on fruit tree navigation task, where the tree depth $d = 6$. Trained on 5000 episodes.

| $N_\omega$ | Execution AQ | | Prediction AQ | | CR F1 | |
|---|---|---|---|---|---|---|
| | Scalarized | Envelope | Scalarized | Envelope | Scalarized | Envelope |
| 1 | $0.7943 \pm 0.039$ | $0.8578 \pm 0.036$ | $0.7153 \pm 0.079$ | $0.6254 \pm 0.041$ | $0.9364 \pm 0.023$ | $0.9706 \pm 0.027$ |
| 4 | $0.8836 \pm 0.01$ | $0.9041 \pm 0.005$ | $0.8195 \pm 0.021$ | $0.7843 \pm 0.055$ | $0.984 \pm 0.016$ | $1 \pm 0$ |
| 8 | $0.8975 \pm 0.003$ | $0.9099 \pm 0.001$ | $0.8427 \pm 0.022$ | $0.7952 \pm 0.023$ | $0.9968 \pm 0.007$ | $1 \pm 0$ |
| 16 | $0.9047 \pm 0.003$ | $0.9109 \pm 0.001$ | $0.8698 \pm 0.01$ | $0.8488 \pm 0.032$ | $1 \pm 0$ | $1 \pm 0$ |
| 32 | $0.9054 \pm 0.003$ | $0.9113 \pm 0.001$ | $0.8647 \pm 0.014$ | $0.8847 \pm 0.006$ | $0.9968 \pm 0.007$ | $1 \pm 0$ |
| 64 | $0.9096 \pm 0.003$ | $0.9119 \pm 0$ | $0.8761 \pm 0.024$ | $0.8731 \pm 0.008$ | $0.9968 \pm 0.007$ | $1 \pm 0$ |
| 128 | $0.9071 \pm 0.004$ | $0.9121 \pm 0$ | $0.8535 \pm 0.033$ | $0.8809 \pm 0.026$ | $1 \pm 0$ | $1 \pm 0$ |

**Table 6:** Sample Efficiency when $d = 5$ - Coverage Ratio (CR) and Adaptation Quality (AQ) comparison of the scalarized and the envelope deep MORL algorithms tested on fruit tree navigation task, where the tree depth $d = 5$. Trained on 5000 episodes.

Table 5 also compares MORL algorithms' adaptation quality in the adaptation phase. Each entry in the table is an average of 5 experiments (train and test). For all numbers of sampled preferences, the envelope deep MORL algorithm has better execution AQ than the scalarized algorithm, in spite of better CR of the envelope version, it also indicates the envelope version algorithm has better alignment between preferences and policies. As $N_\omega$ increases, the values of execution AQ and prediction AQ of both algorithms keep increase. Note that even though when $N_\omega = 1$ the execution AQ of the scalarized version and the envelope version differs only around 0.5, when $N_\omega$ increase to 4, the envelope version algorithm better utilizes the sampled preferences to improve the execution AQ to above 0.9. This agrees with our theoretical analysis that our envelope deep MORL algorithm has better sample efficiency than the scalarized version.

We also investigate how the size of optimality frontiers will affect the performance of our algorithms. We train our deep MORL algorithms on two new FTN environments with $d = 5$ and $d = 7$ respectively. One is smaller than the previous environment, which contains only 32 optimal solutions, the other is larger than the previous environment, containing 128 solutions on the CCS. We fix the number of episode for training as 5000, and test 2000 episode to obtain coverage ratio, and test 5000 episode for policy adaptation quality.

Table 6 shows the results of coverage ratio evaluation in the environment of tree depth $d = 5$. As it shows, both scalarized and envelope algorithms work well in that environment. the CR F1 scores are very close to 1. The envelope version algorithm is more stable than the scalarized version deep MORL algorithm. Besides, the envelope deep MORL algorithm can predict multi-objective solutions, while the scalarized algorithm cannot. The prediction ability will also be improved as $N_\omega$ increases.

**Visualizing Frontiers**    We also provide a visualization of the convex coverage set (CCS) and control frontier for our envelope algorithm and the scalarized baseline in Figure 12. The left figure shows the real CCS and retrieved CCS of both MORL algorithms using t-SNE [50]. We observe that the envelope algorithm (green dots) almost completely covers the entire set of optimal solutions in the real CCS whereas the scalarized algorithm (pink dots) does not. The right figure presents the slices of optimal control frontier and the control frontier of our algorithms along the `Minerals-Water` plane. The envelope version retrieves almost the entire optimal frontier,

| $N_\omega$ | Execution AQ | | Prediction AQ | | CR F1 | |
|---|---|---|---|---|---|---|
| | Scalarized | Envelope | Scalarized | Envelope | Scalarized | Envelope |
| 1 | $0.3748 \pm 0.038$ | $0.6348 \pm 0.063$ | $0.624 \pm 0.012$ | $0.6646 \pm 0.025$ | $0.5847 \pm 0.061$ | $0.60 \pm 0.029$ |
| 4 | $0.5112 \pm 0.063$ | $0.6846 \pm 0.067$ | $0.6704 \pm 0.006$ | $0.730 \pm 0.018$ | $0.6969 \pm 0.057$ | $0.6544 \pm 0.066$ |
| 8 | $0.5376 \pm 0.046$ | $0.7516 \pm 0.039$ | $0.6763 \pm 0.022$ | $0.7237 \pm 0.018$ | $0.6837 \pm 0.097$ | $0.7437 \pm 0.04$ |
| 16 | $0.6052 \pm 0.061$ | $0.7328 \pm 0.043$ | $0.6867 \pm 0.025$ | $0.7473 \pm 0.023$ | $0.6532 \pm 0.029$ | $0.7936 \pm 0.015$ |
| 32 | $0.5646 \pm 0.061$ | $0.7862 \pm 0.049$ | $0.6828 \pm 0.022$ | $0.7204 \pm 0.05$ | $0.6829 \pm 0.037$ | $0.7997 \pm 0.017$ |
| 64 | $0.6168 \pm 0.043$ | $0.743 \pm 0.086$ | $0.671 \pm 0.06$ | $0.7429 \pm 0.015$ | $0.7284 \pm 0.03$ | $0.8112 \pm 0.018$ |
| 128 | $0.6308 \pm 0.058$ | $0.8138 \pm 0.038$ | $0.6916 \pm 0.015$ | $0.7586 \pm 0.024$ | $0.6719 \pm 0.076$ | $0.8199 \pm 0.008$ |

**Table 7:** Sample Efficiency when $d = 7$ - Coverage Ratio (CR) and Adaptation Quality (AQ) comparison of the scalarized and the envelope deep MORL algorithms tested on fruit tree navigation task, where the tree depth $d = 7$. Trained on 5000 episodes.

**Figure 12:** Comparison of CCS and control frontiers of deep MORL algorithms. The left figure (a) is visualizing the real CCS and retrieved CCS of scalarized and envelope MORL algorithms using t-SNE. The right figure (b) presents the slices of optimal control frontier and the control frontier of scalarized and envelope MORL algorithms along the `Mineral-Waters` plane.

while the scalarized version algorithm has larger control discrepancies. The small discrepancy between the real control frontier and the one retrieved by envelope version algorithm at the indentation of the frontier indicates an alignment issue between the preferences and optimal policies.

## C.4 Task-Oriented Dialog Policy Learning (Dialog)

| | Single-(0.5,0.5) | Single-(0.2,0.8) | Single-(0.8,0.2) | Scalarized | Envelope |
|---|---|---|---|---|---|
| SR | $88.18 \pm 0.90$ | $85.30 \pm 0.98$ | $87.62 \pm 0.91$ | $86.38 \pm 0.95$ | $\mathbf{89.52 \pm 0.85}$ |
| #T | $8.93 \pm 0.13$ | $9.40 \pm 0.16$ | $\mathbf{7.42 \pm 0.10}$ | $8.08 \pm 0.12$ | $8.08 \pm 0.12$ |
| UT | $2.13 \pm 0.23$ | $1.84 \pm 0.23$ | $2.53 \pm 0.22$ | $2.38 \pm 0.22$ | $\mathbf{2.65 \pm 0.22}$ |
| AQ | 0.660 | 0.279 | 0.728 | 0.614 | **0.814** |

**Table 8:** The average success rate (SR), number of turns (#T), user utility (UT), and adaptation quality (AQ, $\alpha = 0.1$) of policies obtained by single-objective RL baselines and two MORL algorithms.

Table 8 shows that envelope MORL algorithm achieves best success rate (SR) on average, while the single-objective method with equal weight on both dimensions (0.5) achieves competitive performance. However, on metrics of average utility (UT) and adaptation quality (AQ) (with $\alpha = 0.1$, envelope MORL is significantly better than the other methods, including scalarized MORL. This again demonstrates that adaptability of envelope MORL to tasks with new preferences. The single-objective method with 0.8 success weight has the worst performance on success rate eventually. This is might because lack of turn level reward guidance directly optimize for success is very difficult for the single-objective RL learner.

Figure 13 illustrates utility-weight curves during a policy adaptation phase with given preferences, where each data point is a moving average of closest 500 dialogues in the interval of around $\pm 0.05$ weight of success over three trained policies. We find the envelope deep MORL algorithm is almost always better than other methods in terms of utility under certain given preferences, and the scalarized MORL baseline keeps a good level of

**Figure 13:** Utility-weight curves for the MORL and single-objective RL dialog policy learning after 3,000 training dialogues. We evaluate policies on 5,000 dialogues with near-uniformly randomly sampled preference. For each curve, each data point is a moving average of the closest 500 dialogues in the interval of around ± 0.05 weight of success over three trained policies.

utility under almost all user preferences. Single-objective RL algorithms are good only when the user's weight of success is close to their fixed preferences while training.

## C.5 Multi-Objective SuperMario Game (SuperMario)

| Method | Super Mario | | |
|---|---|---|---|
| | Avg.UT (0.5 x-pos & 0.5 time) | Avg.UT (0.5 coin & 0.5 enemy) | Avg.UT (uniform preference) |
| Scalarized | 317.1 ± 123.7 | 76.7 ± 36.5 | 301.7 ± 49.2 |
| Envelope | **600.9 ± 114.9** | **233.3 ± 31.2** | **319.7 ± 34.4** |

**Table 9:** Performance comparison of different MORL algorithm in learning and adaptation phases on the SuperMario environment under three different preferences. The first preference emphasizes the fast completion of the task so it is 0.5 on x-pos and 0.5 time, the second preference emphasizes collecting coin and eliminating enemy. The third is a uniform preference which has weights 0.2 for all five objectives {x-pos, time, death, coin, enemy}

We compared the average utility of the scalarized baseline and the MORL algorithm with envelope update. The first preference emphasizes the fast completion of the task so it is 0.5 on x-pos and 0.5 time, the second preference emphasizes collecting coin and eliminating enemy. The third is a uniform preference which has weights 0.2 for all five objectives {x-pos, time, death, coin, enemy}. The envelope algorithm outperforms the scalarized algorithm under all these preferences.

**Figure 14:** The training curves of the Envelope Multi-Objective A3C (EMoA3C) algorithm.

**Table 10:** Inferred preferences of the scalarized multi-objective A3C algorithm in different Mario Game variants with 100 episodes.

|    | x-pos | time | life | coin | enemy |
|----|-------|------|------|------|-------|
| g1 | 0.2315 | 0.2569 | 0.335 | 0.0522 | 0.1243 |
| g2 | 0.1600 | 0.1408 | 0.2822 | 0.1955 | 0.2216 |
| g3 | 0.1865 | 0.1837 | 0.0874 | 0.3759 | 0.1665 |
| g4 | 0.1250 | 0.3062 | 0.3522 | 0.1240 | 0.0926 |
| g5 | 0.2378 | 0.1673 | 0.1820 | 0.3432 | 0.0698 |

**Table 11:** The difference between the inferred preferences of the envelope and the scalarized multi-objective A3C algorithms.

|    | x-pos | time | life | coin | enemy |
|----|-------|------|------|------|-------|
| g1 | +0.2973 | -0.0799 | -0.1850 | -0.0052 | -0.0271 |
| g2 | +0.0385 | +0.0829 | -0.0337 | -0.0533 | -0.0348 |
| g3 | +0.0331 | -0.0541 | +0.2667 | -0.1967 | -0.0490 |
| g4 | -0.1039 | -0.0658 | -0.3311 | +0.5720 | -0.0715 |
| g5 | -0.1663 | -0.0635 | +0.0249 | +0.0490 | +0.1555 |

The training curves of EMoA3C algorithm are shown in Figure 14. To plot Figure 14, we use a uniform preference [0.2 0.2 0.2 0.2 0.2] as the probe preference to sample trajectories for evaluation. We observe the algorithm converges within around 5k episodes of training.

Finally we compare the difference between the inferred preferences of the envelope and the scalarized multi-objective A3C algorithm on different variants of SuperMario game, g1 to g5, where only corresponding scalar rewards are available. We only consider maximizing one objective in each game variant, because this orthogonal design helps us characterize the behaviors of agents and compare their inferred preferences. We allow agents to make 100 episodes interactions in each game variance, to determine the preference. Note that only 100 episodes are far from enough for training a single objective reinforcement learning agent, even the model is pre-trained on other tasks.

As Table 11 illustrates, the preferences inferred by the envelope MoA3C agent is more concentrate on the diagonal than that of scalarized MoA3C agent, which is more closer to the true underlying preferences.

Even though the EMoA3C agent can do better, our experiments show that the inferred preferences are not exactly the true underlying preferences. It is mainly for two reasons: First, the trade-off frontiers and policy-preference alignment learned by the algorithm is not ideal. There might be some discrepancies between the obtained control frontier and the real optimal control frontier just like what Figure 12 (b) illustrates. Second, even the agent perfectly learned the frontier and the alignment, close preferences may correspond to the policies with the same expected returns.

We also deploy the trained EMoA3C agents in a Mario Game where only game scores are available. After 100 episodes adaptation, the EMoA3C agent infers that the underlying preference for the achieving higher score mainly focuses on `x-pos` (0.3725) and `time` (0.2307), which is coincident with the strategy human players commonly use – to achieve higher score, especially the stage accomplishment bonus, the first priority is to ensure Mario can move forward towards the flag within the time limit.

## Footnotes

[3]Mario has up to three lives in one episode.