[Reviews · NeurIPS 2019]

Reviewer 1



Summary: The paper proposes a new algorithm for learning linear preferences, which are objectives derived from a linear weighting of a vector reward function, in multi-objective reinforcement learning (MORL). The proposed algorithm achieves this by performing updates that use the convex envelope of the solution frontier to update the parameters of the action-value function, hence its name: envelop Q-learning. This is done by first defining a multi-objective version of the action-value function along with a pseudo-metric, the supremum over the states, actions, and preferences. Then, a Bellman operator is defined for the multi-objective action-value function along with an optimality filter, which together define a new optimality operator. Using all these definitions, the paper then shows three main theoretical results: 1) the optimality operator has a fixed point that maximizes the amount of reward under any given preference, 2) the optimality operator is a contraction, and 3) for any Q in the pseudo-metric space, iterative applications of the optimality operator will result in an action-value function which distance to the fixed point is equal to zero, i.e. is equivalent to the fixed point. Based on these results, the paper then proposes a learning algorithm to approximate the solution of the optimality operator by using a neural network as function approximation, a mixture of two losses to facilitate learning, and double Q-learning to learn the action-value function. Finally, the paper provides empirical evaluations of the envelope Q-learning algorithm in four different domains and demonstrate how it can be used to discover preferences when only scalar rewards are provided. Review: The main idea of the paper is novel and provides interesting avenues for future research. The literature review was thorough and the results seem to include the appropriate baselines and seem to show some improvement over previously proposed algorithms. However, I have two main concerns about the results presented in the paper. First, the lack of clarity and details in the background and theoretical proofs, which makes it difficult for the reader to verify the results. In other words, I could not verify that the theoretical results were correct and, although I do not think they are wrong, I could not rule out the possibility that there was a mistake. I think this is mostly because some parts of the proofs skip over too many steps and not because the proofs are wrong. Second, the empirical results presented seem to be based on one single run which was tested over several examples. If this is not the case, the paper should include details about the number of runs used to compute the statistics; note that the number of runs is not equivalent to the number of episodes or training sessions. However, if this is the case, then it is problematic because the performance of deep reinforcement learning agents can vary significantly even when using the same initialization scheme. For all these reasons I consider the paper should be rejected but only marginally below the acceptance threshold. I provide more detailed comments in the following sections. On the clarity of the introduction: Overall, the introduction was very clear and well motivated except for two exceptions: 1. In line 32, the statement: “The MORL framework provides two distinct advantages — (1) reduced dependence on reward design,” is not clear. Since in MORL the reward function is a vector and tasks are defined as a function of a preference, it seems that designers would have to define a more complicated reward function or a general reward function but a more complicated preference function. Hence, it is not clear from that statement alone that there’s reduced dependence on reward design. Perhaps I’m missing something, but to improve the clarity, it would be good if more evidence was provided to support this claim. 2. In lines 32 to 34, 46 to 48, and 49 to 50, there are three independent numbered lists with two items each. I personally found this unnecessary and confusing since it’s not easy to keep track of all these lists and it is not clear what is the purpose of each of them. This is mostly a minor complaint and a stylistic choice, but I think it would improve the clarity of the paper if you only had one list to emphasize the key insights of your approach (lines 49 to 50). These are mostly minor observations, but I think addressing them would improve the quality of the paper. On the clarity of the background: My main concern about the background section is the lack of distinction between the Pareto Frontier and the Convex Coverage Set (CCS in the equation below line 72), which is only exacerbated by the lack of details provided for the plots in Figure 2. My first suggestion for addressing this issue is to have a formal definition for the Pareto Frontier (F* in the paper in line 71) under the MORL framework and emphasize how it is different from the CCS. Second, to improve the plots in Figure 2 on page 3, the paper should provide more context about what each element of the plot and each label means. Moreover, consider using different colors instead of a grayscale. Third, the plot should exclude irrelevant information, for example the circles labeled L and K in Figure 2(a) and the dashed lines in Figure 2(b). Lastly, there was no context or details provided for Figure 2(c), so I would suggest either removing it or providing some explanation for it. There’s some allusion to the meaning of the plot in Figure 2(c) in line 115, but it is not enough information to understand the plot. On the clarity of the theoretical results: 1. First, I would suggest defining the multi-objective optimality filter H in a separate line since it is an important definition. 2. Second, I could not make sense of the proof provided for Theorem 1 even after several attempts at it. Specifically, in the equations below line 601 in Appendix A.2.1, it is not clear how ⍵^T and the arg_Q operator cancel each other. Additionally, the Q in this same set of equations should be labeled Q* since the whole proof is based on the optimal multi-objective action-value function. Moreover, I believe that in the equation below line 604, there is an arg_Q operator missing before the supremum operator; otherwise, the equality would not hold. Finally, in the same equation, two different symbols are used to denote the transpose operator, which is confusing. 3. Third, just as in Theorem 1, the proof for Theorem 2 is also missing details. In the set of equations below line 614, it is not clear how ⍵^T cancels out from line 3 to line 4. Allegedly, this is because some cancellation between ⍵^T and the arg_Q operator, but, once again, it is not clear how this happens. Lastly, in the set of equations below line 616, it is not clear how line 3 follows from line 4. This does not mean that I think the proofs are wrong, but I cannot verify that they are true. It would be great if you could provide a more detailed derivation for the less mathematically inclined readers. 4. Finally, in the learning algorithms part of Section 3 (Lines 169 to 202), it would be useful if the paper provided some intuition about why L^B is smoother than L^A. Moreover, there is a mismatch between the notation used for the losses in the main body of the paper and the notation used in the appendices. In the main paper, the losses are denoted L^A and L^B (except for one occurrence in line 190 where L^B is referred to as L^B_k), whereas in the appendices these loses are denoted L^A_k and L^B_k. To avoid confusion, the paper should keep the notation consistent, including in the appendices. On the significance and clarity of the empirical evaluations: 1. First, the plots in Figure 3 shows similar issues as the plots in Figure 2: not everything is well defined and not enough context is provided about how the main body relates to the plots in the figure. 2. Second, no information is provided about the number of runs used for the results. There is information about the number of test episodes, but given that deep reinforcement agents can have drastically different performance based on the random initialization, several different independent runs should be used in order to demonstrate meaningful differences in performance. Moreover, there is no information about the meaning of the error bars. If computational resources are a limitation, consider providing statistically meaningful results for the smaller environments and just a few runs (e.g., 5) for the bigger environments. ---- Update: Given that the authors addressed all my main concerns, I have decided to increase my overall score. I still think that Figure 2 and 3 require a more detailed explanation and hope that the authors will address this in the final version of the paper.

Reviewer 2



=== UPDATE === After reading the author response, etc., I do agree with reviewer 1 and think the proofs present in the author response makes the math easier to follow (although I was able to do it previously with a non-negligible amount of effort). Regarding Minecart vs. experiments in this paper, I still don't have an intuitive feel for the significance of the empirical results (as I don't specifically work in the MORL space), but I was convinced in the first place. I still think it is a good paper that I would prefer to see accepted. ====== The authors consider the problem of multi-objective reinforcement learning (MORL), proposing a solution that learns a single policy network over the envelope of the convex coverage set of the Pareto frontier wrt a linear combination of preferences. This is an innovation over scalarized deep Q-learning [Mossalam, et al., 2016] (and follow-up work [Abels, et al., ICML19]) in that this previous work sequentially trains Q-networks on corner weights in preference space (thus approximating the CCS), resulting in a set of Q-networks (as opposed to a single-network policy). This is basically accomplished by performing (deep) Q-Learning over a generalized Bellman equation that directly utilizes vectorized rewards such that they generalize to all weight parameters in the CCS — using hindsight replay and homotopy optimization to make learning tractable (and presumably stable). Once this generalized setting is learned, it immediately adapts to any set of weight parameters as a deep learner is learning a function approximation over the entire space. To accommodate scalarizations of the multi-objective problem in the reward, they also introduce a method for learning the weights. Analytical analysis mostly centers around convergence of the generalized Bellman equations and that weight parameters selected from this function are equivalent to learning the single preference function directly. Experiments are conducted on four different domains (including simple and challenging tasks), demonstrating empirical improvements in terms of coverage ratio (ability to recover optimal policies), adaptation error (difference between recovered control frontier and optimal one), and average utility over recent relevant works (including [Abels, et al., ICML19]). Overall, this seems another step in the deep learning evolution to formulate the problem directly (in this case, by generalizing the Bellman equations) and relying on the expressivity of the function approximator to directly learn the function from data. On one hand, it is sort of surprising it works as one would think that the generalized Bellman equations would be difficult to learn and require really massive datasets and potentially finicky optimization methods (i.e., more than hindsight replay and homotopic optimization — as is sometimes seen the Appendix details) to get this to work. However, I suppose since the CCS is relatively smooth, maybe this isn’t a big issue with the right architecture. Conceptually, it is an appealing framework, has sufficient analytical analysis to demonstrate depth of understanding, and the empirical results are reasonably compelling. Thus, while noting that I have little experience with MORL (and much more with RL in general) and that I do have some criticisms as itemized below, I think this paper is an improvement over recent related works and will likely be well-received and used within this community. Below, I will evaluate this submission along the requisite dimensions. === Originality Obviously, MORL is a well-established problem — however the conceptual approach of using the vectorized values directly and generalizing the Bellman equations to derive a single-network solution as opposed to scalarizing the vectorized values and learning multiple networks for different configurations seems a non-negligible step that is more elegant and greater runway for further innovations. Specifically, it is an important distinction that learning the boundaries of the CCS may not generalize well to arbitrary weight parameters at test time as is done in the multi-network case. The solution proposed in this work will almost certainly lead to a better interpolation of unseen weight configurations and the theory shows that it will return optimal Q-networks when actually on the CCS envelope. As I previously state, I am a bit surprised it actually works due to the expressivity of the resulting function, but I was also surprised when first seeing DQN work as well as it does in general, but wish I had more time to really dig into the code to see the details. However, I will view that I want to do this as a positive. Wrt originality, I believe it meets the NeurIPS publication bar. === Quality The general framework make sense and the theoretical analysis supports it well. For the ‘inverse weight parameters’ problem, it is not entirely clear the MVN assumption of weights makes sense, but seems to work and is fine for an introduction of the framework. Something like a GP or other bayesian optimization procedure seems to make more sense, but this can be follow-up work and isn’t the core contribution of the paper. The empirical results, while sufficiently convincing, offer more room for improvement. Specifically, [Abels, et al. ICML19] use a minecart simulation which seems to be relatively difficult and since this paper makes a direct comparison in general, I am not certain why this wasn’t used. Deep Sea Treasure and Fruit Tree Navigation don’t seem particularly difficult, while dialogue management and SuperMario do. Some discussion here regarding the relative difficulty, convergence times, etc. would be nice. Specific to [Abels, et al., ICML19], while I don’t know of the a better comparison, this isn’t a perfect match as this paper was really about adaptation under non-stationary weights — even if they do use the multi-network formulation. Some discussion in this regard would also be nice. Wrt quality, I also believe it meets the publication bar, although more due to concept and theory than empirical results and not as strongly as originality. === Clarity For somebody very familiar with RL in general, the paper is easy to follow and reasonably well self-contained. That being said, the Appendices do add quite a bit of material that makes the paper more understandable — but do believe the paper stands as relatively complete on its own. My one conceptual question is lines 44-45, “optimized over the entire space of preferences in a domain” vs. lines 166-168, “same utilities…will only differ when the utility corresponds to a recess in the frontier”. By the definition of CCS, the second statement is true (along with lines 81-82) — is the first statement just referring to the expressivity of the learned function? The construct used for training will certainly bias the portion of the space that is learned. A few small questions regarding details; what were the values of hyperparameters N_\tau and N_\omega in Algorithm 1 and what was the rate of annealing for homotopy optimization? In any case, this is a relatively simple idea (once revealed) and well-motivated. The theoretical analysis is sufficient and the experiments are well detailed (even better once the Appendices are read). === Significance I believe the overall conceptual framework is sufficiently significant, the theory is sufficiently well-supported, and there is enough evidence that they actually got this to work. I didn’t spend a ton on time looking over the code and didn’t run, but it ‘appears’ usable just from skimming it. The least well-studied contribution is the ‘recovering linear weights’ component and the ‘2x average improvement’ of line 61 does show preferences can be learned, but is a bit of a straw man on the whole — but this isn’t the core contribution of the paper. Overall, while I am not of this specific community, I think the overall framework is elegant and shown to work. I would expect it to be the starting point for work in this area as it handles vector rewards directly and is conceptually very elegant — fitting into the continuing trend of general frameworks with lots of data.

Reviewer 3



***** Post-rebuttal comments ***** Thank you for addressing my queries in your response. I think including a discussion on the contributions over and above Abels et al (2019), as well as the other issues raised, will help improve the final version of the paper. Summary: The paper presents a method for learning a preference-conditioned Q value function that can quickly adapt and solve new tasks with different preferences (in the case of linear preferences). A generalised Bellman operator is introduced, and convergence to optimal Q values is proved. This allows for the approach to learn a single Q function over all preferences. At test time, dynamic preferences can be fed into the function to produce the optimal value functions for the corresponding task. Results in both low and high-dimensional domains validate the method, which improves over existing work. I think the paper is extremely well written and the idea as a whole to be the "right" way of tackling the problem. I only have some concerns in terms of how big a contribution it is compared to previous work. Comments: I found the paper to be extremely well-written, easy to read and clear in its explanations. I thought the algorithm block was very useful and informative, and I really liked the Super Mario domain. I think Figure 10a is also very useful and authors may wish to consider including it in the main text, space permitting. The only comment I have on clarity is about Figure 2c. It's not quite clear what is going in on the diagram and how it links to (c)'s caption. Which solutions do scalarised algorithms find in this diagram? In particular, the claim on L115 says that the envelope method leads to better learning as demonstrated in the figure, but it's not clear to me how. My main concern revolves around the contributions and improvements offered by the paper, in particular compared to Abels et al. The paper claims that only that "scalarized updates are not sample efficient and lead to sub-optimal MORL policies", but I think a more detailed explanation should be provided, especially since they appear to share many similarities. Do the main differences come down to a lack of convergence/optimality guarantees? It seems like the idea of learning a preference-conditioned Q function is that same here as it was in that previous work. Additionally, while the results show the method outperforming the baselines, I find it hard to judge how big an improvement it is. Certainly compared to scalarised, the numbers in Table 1 look close together (the utility in particular), and Figure 13, for example, looks like extremely minimal gains. As such, I struggle to judge whether the improvements are significant or incremental. Including some significance testing on the results would be helpful here. The paper claims "And to achieve the same level AE the envelope algorithm requires smaller N_omega than the scalarized algorithm. This reinforces our theoretical analysis that the envelope MORL algorithm has better sample efficiency than the scalarized version." I don't understand this paragraph for two reasons. It suggests that there is theory that shows that the sample complexity is lower than that of the scalarised version, but I don't see any theory suggesting that (unless I am missing something). Additionally, picking a larger N_omega has no effect on the number of interactions with the environment, since it is just more omegas being applied to the same (s, a, r, s') transition. Could the authors please clarify this paragraph? I have a question regarding the loss functions used when learning. L^A and L^B are defined, where L^A is the standard loss function and L^B is a surrogate one, and the final loss is a mixture of the two. Given that L^A is the actual DQN loss, why is lambda annealed toward L^B instead of L^A? Intuitively, wouldn't you want L^A to be the final loss optimised, and so anneal lambda in the other direction? In Algorithm 1, the weights are sampled and y_{ij} is calculated. In that section it says that omega prime is an element of W. What is W? One separate area of research that may be worth mentioning in relation to this work is composition. There's been some recent work on composing value functions and policies to solve new tasks [1, 2, 3, 4]. In this setting, you could imagine learning separate policies for each objective individually, and the composing them to solve the multi-objective task, where the weights of the composed policy are related to the preferences. Of course this has drawbacks in terms of sample and memory complexity, but it may still be worth discussing. The link to code that was provided returned "This upload does not exist", and so I was unable to review the code. Minor corrections: At the risk of being overly pedantic, mention that s' comes from the trajectory tau in Eq 3. Also, the discount factor is only introduced on L158, but is used much earlier (obviously gamma as the discount factor is common knowledge, but it would still be good to mention it around Eq 1. L136: omega should be bold Caption table 2: learnig -> learning [1] Todorov, Emanuel. "Compositionality of optimal control laws." Advances in Neural Information Processing Systems. 2009. [2] Hunt, Jonathan J., et al. "Composing Entropic Policies using Divergence Correction." International Conference on Machine Learning. 2019. [3] Van Niekerk, Benjamin, et al. "Composing Value Functions in Reinforcement Learning." International Conference on Machine Learning. 2019. [4] Peng, Xue Bin, et al. "MCP: Learning Composable Hierarchical Control with Multiplicative Compositional Policies." arXiv preprint arXiv:1905.09808 (2019)

[Author Response · NeurIPS 2019]

Thank you for the constructive comments and suggestions. We will incorporate all presentation improvements suggested.

**Theoretical results (Reviewer 1):** [$\boldsymbol{\omega}^{\mathsf{T}}$ and $\arg_Q$ cancel each other] By definition (line 143),
$\arg_Q \sup_{a \in \mathcal{A}, \boldsymbol{\omega}' \in \Omega} \boldsymbol{\omega}^{\mathsf{T}} \boldsymbol{Q}(s, a, \boldsymbol{\omega}') := \boldsymbol{Q}(s, a', \boldsymbol{\omega}'')$, i.e., the $\arg_Q$ operator extracts the $\boldsymbol{Q}$ value that results in the largest
utility using the preferences $\boldsymbol{\omega}$. Therefore, linearizing this $\boldsymbol{Q}$ with the same $\boldsymbol{\omega}$ results in exactly the same supremum, i.e.,
$\boldsymbol{\omega}^{\mathsf{T}} \arg_Q \sup_{a \in \mathcal{A}, \boldsymbol{\omega}' \in \Omega} \boldsymbol{\omega}^{\mathsf{T}} \boldsymbol{Q}(s, a, \boldsymbol{\omega}') = \boldsymbol{\omega}^{\mathsf{T}} \boldsymbol{Q}(s, a', \boldsymbol{\omega}'') = \sup_{a \in \mathcal{A}, \boldsymbol{\omega}' \in \Omega} \boldsymbol{\omega}^{\mathsf{T}} \boldsymbol{Q}(s, a', \boldsymbol{\omega}')$. **Note the supremum is over $\boldsymbol{\omega}'$, not $\boldsymbol{\omega}$.**

[Theorem 1] Thanks for catching the typo - $\boldsymbol{Q}$ should be $\boldsymbol{Q}^*$. We realize that the proofs are a bit compressed - we will
update the paper with more detailed derivations for all proofs. Here is Thm. 1 in detail (starting step 2 under line 601):

$$\boldsymbol{\omega}^{\mathsf{T}} \mathcal{T} \boldsymbol{Q}^*(s, a, \boldsymbol{\omega}) = \boldsymbol{\omega}^{\mathsf{T}} \boldsymbol{r}(s, a) + \gamma \cdot \boldsymbol{\omega}^{\mathsf{T}} \mathbb{E}_{s' \sim \mathcal{P}(\cdot|s,a)} \arg_Q \sup_{a' \in \mathcal{A}, \boldsymbol{\omega}' \in \Omega} \boldsymbol{\omega}^{\mathsf{T}} \boldsymbol{Q}^*(s', a', \boldsymbol{\omega}')$$

$$\text{(linearity of exp. \& cancel } \boldsymbol{\omega}^{\mathsf{T}} \text{ and } \arg_Q) = \boldsymbol{\omega}^{\mathsf{T}} \boldsymbol{r}(s, a) + \gamma \cdot \mathbb{E}_{s' \sim \mathcal{P}(\cdot|s,a)} \sup_{a' \in \mathcal{A}, \boldsymbol{\omega}' \in \Omega} \boldsymbol{\omega}^{\mathsf{T}} \boldsymbol{Q}^*(s', a', \boldsymbol{\omega}')$$

$$\text{(insert eq. (20), def. of } \boldsymbol{Q}^*) = \boldsymbol{\omega}^{\mathsf{T}} \boldsymbol{r}(s, a) + \gamma \cdot \mathbb{E}_{s' \sim \mathcal{P}(\cdot|s,a)} \sup_{a' \in \mathcal{A}, \boldsymbol{\omega}' \in \Omega} \boldsymbol{\omega}^{\mathsf{T}} \left\{ \arg_Q \sup_{\pi \in \Pi} \boldsymbol{\omega}'^{\mathsf{T}} \mathbb{E}_{\tau \sim (\mathcal{P}, \pi) \atop |s_0 = s', a_0 = a'} \left[ \sum_{t=0}^{\infty} \gamma^t \boldsymbol{r}(s_t, a_t) \right] \right\}$$

$$\text{(use def. of } \arg_Q, \text{ explained below)} = \boldsymbol{\omega}^{\mathsf{T}} \boldsymbol{r}(s, a) + \gamma \cdot \mathbb{E}_{s' \sim \mathcal{P}(\cdot|s,a)} \sup_{a' \in \mathcal{A}} \boldsymbol{\omega}^{\mathsf{T}} \left\{ \arg_Q \sup_{\pi \in \Pi} \boldsymbol{\omega}^{\mathsf{T}} \mathbb{E}_{\tau \sim (\mathcal{P}, \pi) \atop |s_0 = s', a_0 = a'} \left[ \sum_{t=0}^{\infty} \gamma^t \boldsymbol{r}(s_t, a_t) \right] \right\}$$

$$\text{(rearrange expectation and sup)} = \boldsymbol{\omega}^{\mathsf{T}} \boldsymbol{r}(s, a) + \gamma \cdot \boldsymbol{\omega}^{\mathsf{T}} \arg_Q \sup_{\pi \in \Pi} \boldsymbol{\omega}^{\mathsf{T}} \mathbb{E}_{\tau \sim (\mathcal{P}, \pi) \atop s_0 \sim \mathcal{P}(\cdot|s,a)} \left[ \sum_{t=0}^{\infty} \gamma^t \boldsymbol{r}(s_t, a_t) \right]$$

$$\text{(merge 1st term to sum \& use def. of } \boldsymbol{Q}^* \text{ again)} = \boldsymbol{\omega}^{\mathsf{T}} \left\{ \arg_Q \sup_{\pi \in \Pi} \boldsymbol{\omega}^{\mathsf{T}} \mathbb{E}_{\tau \sim (\mathcal{P}, \pi) \atop |s_0 = s, a_0 = a} \left[ \sum_{t=0}^{\infty} \gamma^t \boldsymbol{r}(s_t, a_t) \right] \right\} = \boldsymbol{\omega}^{\mathsf{T}} \boldsymbol{Q}^*(s, a, \boldsymbol{\omega})$$

The fourth equation is due to a sandwich inequality, $\boldsymbol{\omega}^{\mathsf{T}} \arg_Q \sup_{\pi \in \Pi} \boldsymbol{\omega}^{\mathsf{T}} \boldsymbol{Q}^{\pi} \leq \sup_{\boldsymbol{\omega}' \in \Omega} \boldsymbol{\omega}^{\mathsf{T}} \arg_Q \sup_{\pi \in \Pi} \boldsymbol{\omega}'^{\mathsf{T}} \boldsymbol{Q}^{\pi} = \boldsymbol{\omega}^{\mathsf{T}} \arg_Q \sup_{\pi \in \Pi} \boldsymbol{\omega}_*^{\mathsf{T}} \boldsymbol{Q}^{\pi} =$

$\boldsymbol{\omega}^{\mathsf{T}} \boldsymbol{Q}^{\pi'_{\boldsymbol{\omega}'_*}} \leq \boldsymbol{\omega}^{\mathsf{T}} \arg_Q \sup_{\pi \in \Pi} \boldsymbol{\omega}^{\mathsf{T}} \boldsymbol{Q}^{\pi}$, where $\boldsymbol{\omega}'_*$ and $\pi'_{\boldsymbol{\omega}'_*}$ are preference and policy corresponding to the supremums.

[Theorem 2] Step 2 to 3 (line 614) is because $|\mathbb{E}[\cdot]| \leq \mathbb{E}[|\cdot|] \leq \sup|\cdot|$, and step 3 to 4 results from the cancellation
between $\boldsymbol{\omega}^{\mathsf{T}}$ and $\arg_Q$ (as justified above). After line 616, step 2 to 3 arises from the w.l.o.g. assumption that
$\boldsymbol{\omega}^{\mathsf{T}} \boldsymbol{Q}(s', a', \boldsymbol{\omega}') - \sup_{a'', \boldsymbol{\omega}''} \boldsymbol{\omega}^{\mathsf{T}} \boldsymbol{Q}'(s', a'', \boldsymbol{\omega}'') \geq 0$, as stated in lines 612 and 615. Thus, the whole expression in $|\cdot|$ is
nonnegative and $\boldsymbol{\omega}^{\mathsf{T}} \boldsymbol{Q}(s', a', \boldsymbol{\omega}') - \boldsymbol{\omega}^{\mathsf{T}} \boldsymbol{Q}'(s', a', \boldsymbol{\omega}') \geq 0$. We can discard the last two terms since $\boldsymbol{\omega}^{\mathsf{T}} \boldsymbol{Q}'(s', a', \boldsymbol{\omega}') \leq$
$\sup_{a'' \in \mathcal{A}, \boldsymbol{\omega}'' \in \Omega} \boldsymbol{\omega}^{\mathsf{T}} \boldsymbol{Q}'(s', a'', \boldsymbol{\omega}'')$. Step 3 to 4 is because $\sup_{s', \boldsymbol{\omega}'} f(s', a', \boldsymbol{\omega}') \leq \sup_{s', a'', \boldsymbol{\omega}'} f(s', a'', \boldsymbol{\omega}')$ holds for any $a'$ and $f(\cdot)$.

**Empirical results (Reviewers 1 and 3):** [Multiple runs and error bars] Each data point in Table 1 indicates the **mean**
**and standard deviation** over **5 independent** training and test runs, for all methods in all four domains. The error bars
in Figure 4 are standard deviations of CR and AE estimated from 5 independent runs under each configuration. This is
mentioned in lines 228, 860-862, 877, 881, but we will consolidate and make this clearer for the reader.

[Statistical tests] We performed the unpaired t-test between our envelope model and the baselines and achieved
significance scores of $p < 0.05$ vs MOFQI on all domains, $p < 0.01$ vs CN+OLS on DST and $p < 0.05$ vs Scalarized
on FTN, Dialog and SuperMario. We will add this information to the results table.

**Comparison with Abels, et al. (Reviewers 2 and 3):** There are 3 key contributions that distinguish our work from
Abels et. al., 2019. We will add a better description of these to the paper as well as better explain figures 2 & 3.

[Algorithmic] Our algorithm (envelope Q-learning), utilizes the convex envelope of the solution frontier to update
parameters of the policy network, using an optimality filter $\mathcal{H}$ (line 142) which maintains $\sup_{\boldsymbol{\omega}'} \boldsymbol{\omega}^{\mathsf{T}} Q(\cdot, \cdot, \boldsymbol{\omega}')$. This
allows our method to quickly align one preference with optimal rewards and trajectories that may have been explored
under other preferences. Abels et al. on the other hand, use scalarized updates that optimizes the scalar utility and hence
cannot use the information of $\max_a Q(s, a, \boldsymbol{\omega}')$ to update the optimal solution aligned with a different $\boldsymbol{\omega}$. As illustrated
in Figure 2 (c), assuming we have found two optimal solutions D and F in the CCS, misaligned with preferences $\boldsymbol{\omega}_2$ and
$\boldsymbol{\omega}_1$. The scalarized update cannot use the information of $\max_a Q(s, a, \boldsymbol{\omega}_1)$ (corresponding to F) to update the optimal
solution aligned with $\boldsymbol{\omega}_2$ or vice versa. It only searches along $\boldsymbol{\omega}_1$ direction leading to non-optimal L, even if solution D
has been seen under $\boldsymbol{\omega}_2$. Hence, our algorithm has better sample efficiency, as is also seen from the empirical results.

[Theoretical] Further, we introduce a **theoretical framework** for designing and analyzing value-based MORL al-
gorithms, and **convergence proofs** for our envelope Q-learning algorithm. Abels et al., whose method can also be
analyzed under our framework, do not provide theoretical analyses of the correctness or convergence of their algorithm.

[Empirical] We also provide **new evaluation metrics and benchmark environments for MORL** – CR and AE. In
terms of experiments, Abels et al. only evaluate on two synthetic domains – DST and Minecart. We apply our algorithm
to a wider variety of domains including DST, FTN and **two complex larger scale domains** – task-oriented dialog and
supermario. Our FTN domain (128 solutions) is a scaled up, more complex version of Minecart ($< 10$ solutions).

[Meta-Review · NeurIPS 2019]

all three reviewers agree the paper provide some novel and significant contribution and would like to see the paper presented in the conference.